# Improved Autoregressive Modeling with Distribution Smoothing

**Chenlin Meng, Jiaming Song, Yang Song, Shengjia Zhao & Stefano Ermon**
Stanford University
{chenlin,tsong,yangsong,sjzhao,ermon}@cs.stanford.edu

## Abstract

While autoregressive models excel at image compression, their sample quality is often lacking. Although not realistic, generated images often have high likelihood according to the model, resembling the case of adversarial examples. Inspired by a successful adversarial defense method, we incorporate randomized smoothing into autoregressive generative modeling. We first model a smoothed version of the data distribution, and then reverse the smoothing process to recover the original data distribution. This procedure drastically improves the sample quality of existing autoregressive models on several synthetic and real-world image datasets while obtaining competitive likelihoods on synthetic datasets.

## 1 Introduction

Autoregressive models have exhibited promising results in a variety of downstream tasks. For instance, they have shown success in compressing images (Minnen et al., 2018), synthesizing speech (Oord et al., 2016a) and modeling complex decision rules in games (Vinyals et al., 2019). However, the sample quality of autoregressive models on real-world image datasets is still lacking.

Poor sample quality might be explained by the manifold hypothesis: many real world data distributions (*e.g.* natural images) lie in the vicinity of a low-dimensional manifold (Belkin & Niyogi, 2003), leading to complicated densities with sharp transitions (*i.e.* high Lipschitz constants), which are known to be difficult to model for density models such as normalizing flows (Cornish et al., 2019). Since each conditional of an autoregressive model is a 1-dimensional normalizing flow (given a fixed context of previous pixels), a high Lipschitz constant will likely hinder learning of autoregressive models.

Another reason for poor sample quality is the "compounding error" issue in autoregressive modeling. To see this, we note that an autoregressive model relies on the previously generated context to make a prediction; once a mistake is made, the model is likely to make another mistake which compounds (Kääriäinen, 2006), eventually resulting in questionable and unrealistic samples. Intuitively, one would expect the model to assign low-likelihoods to such unrealistic images, however, this is not always the case. In fact, the generated samples, although appearing unrealistic, often are assigned high-likelihoods by the autoregressive model, resembling an "adversarial example" (Szegedy et al., 2013; Biggio et al., 2013), an input that causes the model to output an incorrect answer with high confidence.

Inspired by the recent success of randomized smoothing techniques in adversarial defense (Cohen et al., 2019), we propose to apply randomized smoothing to autoregressive generative modeling. More specifically, we propose to address a density estimation problem via a two-stage process. Unlike Cohen et al. (2019) which applies smoothing to the model to make it more robust, we apply smoothing to the data distribution. Specifically, we convolve a symmetric and stationary noise distribution with the data distribution to obtain a new "smoother" distribution. In the first stage, we model the smoothed version of the data distribution using an autoregressive model. In the second stage, we reverse the smoothing process—a procedure which can also be understood as "denoising"—by either applying a gradient-based denoising approach (Alain & Bengio, 2014) or introducing another conditional autoregressive model to recover the original data distribution from the smoothed one. By choosing an appropriate smoothing distribution, we aim to make each step easier than the original learning problem: smoothing facilitates learning in the first stage by making the input distribution

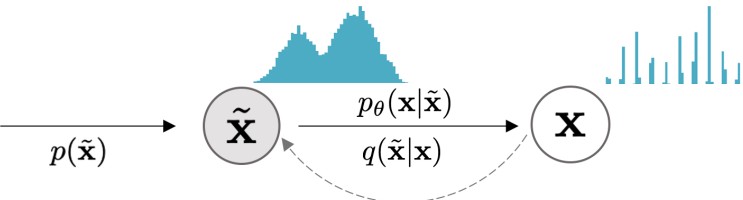

Figure 1: Overview of our method. From a data distribution ($\mathbf{x}$) we inject noise ($q(\tilde{\mathbf{x}}|\mathbf{x})$) which makes the distribution smoother ($\tilde{\mathbf{x}}$); then we model the smoothed distribution ($p_\theta(\tilde{\mathbf{x}})$) as well as the denoising step ($p_\theta(\mathbf{x}|\tilde{\mathbf{x}})$), forming a two-step model.

fully supported without sharp transitions in the density function; generating a sample given a noisy one is easier than generating a sample from scratch.

We show with extensive experimental results that our approach is able to drastically improve the sample quality of current autoregressive models on several synthetic datasets and real-world image datasets, while obtaining competitive likelihoods on synthetic datasets. We empirically demonstrate that our method can also be applied to density estimation, image inpainting, and image denoising.

## 2 BACKGROUND

We consider a density estimation problem. Given $D$-dimensional i.i.d samples $\{\mathbf{x}_1, \mathbf{x}_2, ..., \mathbf{x}_N\}$ from a continuous data distribution $p_{\text{data}}(\mathbf{x})$, the goal is to approximate $p_{\text{data}}(\mathbf{x})$ with a model $p_\theta(\mathbf{x})$ parameterized by $\theta$. A commonly used approach for density estimation is maximum likelihood estimation (MLE), where the objective is to maximize $L(\theta) \triangleq \frac{1}{N} \sum_{i=1}^{N} \log p_\theta(\mathbf{x}_i)$.

### 2.1 AUTOREGRESSIVE MODELS

An autoregressive model (Larochelle & Murray, 2011; Salimans et al., 2017) decomposes a joint distribution $p_\theta(\mathbf{x})$ into the product of univariate conditionals:

$$p_\theta(\mathbf{x}) = \prod_{i=1}^{D} p_\theta(x_i|\mathbf{x}_{<i}), \tag{1}$$

where $x_i$ stands for the $i$-th component of $\mathbf{x}$, and $\mathbf{x}_{<i}$ refers to the components with indices smaller than $i$. In general, an autoregressive model parameterizes each conditional $p_\theta(x_i|\mathbf{x}_{<i})$ using a pre-specified density function (*e.g.* mixture of logistics). This bounds the capacity of the model by limiting the number of modes for each conditional.

Although autoregressive models have achieved top likelihoods amongst all types of density based models, their sample quality is still lacking compared to energy-based models (Du & Mordatch, 2019) and score-based models (Song & Ermon, 2019). We believe this can be caused by the following two reasons.

### 2.2 MANIFOLD HYPOTHESIS

Several existing methods (Roweis & Saul, 2000; Tenenbaum et al., 2000) rely on the manifold hypothesis, *i.e.* that real-world high-dimensional data tends to lie on a low-dimensional manifold (Narayanan & Mitter, 2010). If the manifold hypothesis is true, then the density of the data distribution is not well defined in the ambient space; if the manifold hypothesis holds only approximately and the data lies in the vicinity of a manifold, then only points that are very close to the manifold would have high density, while all other points would have close to zero density. Thus we may expect the data density around the manifold to have large first-order derivatives, i.e. the density function has a high Lipschitz constant (if not infinity).

To see this, let us consider a 2-d example where the data distribution is a thin ring distribution (almost a unit circle) formed by rotating the 1-d Gaussian distribution $\mathcal{N}(1, 0.01^2)$ around the origin. The density function of the ring has a high Lipschitz constant near the "boundary". Let us focus on a data point travelling along the diagonal as shown in the leftmost panel in figure 2. We plot the first-order

Figure 2: Manifold hypothesis illustration. The data point is travelling along the diagonal as shown in the leftmost panel. The white arrow stands for the direction and magnitude of the derivative of density at the data point. The data location for each figure is $(\sqrt{0.5} + c, \sqrt{0.5} + c)$, where $c$ is the number below each figure and $(\sqrt{0.5}, \sqrt{0.5})$ is the upper right intersection of the trajectory with the unit circle.

directional derivatives of the density for the point as it approaches the boundary from the inside, then lands on the ring, and finally moves outside the ring (see figure 2). As we can see, when the point is far from the boundary, the derivative has a small magnitude. When the point moves closer to the boundary, the magnitude increases and changes significantly near the boundary even with small displacements in the trajectory. However, once the point has landed on the ring, the magnitude starts to decrease. As it gradually moves off the ring, the magnitude first increases and then decreases just like when the point approached the boundary from the inside. It has been observed that certain likelihood models, such as normalizing flows, exhibit pathological behaviors on data distributions whose densities have high Lipschitz constants (Cornish et al., 2019). Since each conditional of an autoregressive model is a 1-d normalizing flow given a fixed context, a high Lipschitz constant on data density could also hinder learning of autoregressive models.

## 2.3  COMPOUNDING ERRORS IN AUTOREGRESSIVE MODELING

Autoregressive models can also be susceptible to compounding errors from the conditional distributions (Lamb et al., 2016) during sampling time. We notice that an autoregressive model $p_\theta(\mathbf{x})$ learns the joint density $p_{\text{data}}(\mathbf{x})$ by matching each of the conditional $p_\theta(x_i|\mathbf{x}_{<i})$ with $p_{\text{data}}(x_i|\mathbf{x}_{<i})$. In practice, we typically have access to a limited amount of training data, which makes it hard for an autoregressive model to capture all the conditional distributions correctly due to the curse of dimensionality. During sampling, since a prediction is made based on the previously generated context, once a mistake is made at a previous step, the model is likely to make more mistakes in the later steps, eventually generating a sample $\hat{\mathbf{x}}$ that is far from being an actual image, but is mistakenly assigned a high-likelihood by the model.

The generated image $\hat{\mathbf{x}}$, being unrealistic but assigned a high-likelihood, resembles an adversarial example, i.e., an input that causes the model to make mistakes. Recent works (Cohen et al., 2019) in adversarial defense have shown that random noise can be used to improve the model's robustness to adversarial perturbations — a process during which adversarial examples that are close to actual data are generated to fool the model. We hypothesize that such approach can also be applied to improve an autoregressive modeling process by making the model less vulnerable to compounding errors occurred during density estimation. Inspired by the success of randomized smoothing in adversarial defense (Cohen et al., 2019), we propose to apply smoothing to autoregressive modeling to address the problems mentioned above.

## 3  GENERATIVE MODELS WITH DISTRIBUTION SMOOTHING

In the following, we propose to decompose a density estimation task into a smoothed data modeling problem followed by an inverse smoothing problem where we recover the true data density from the smoothed one.

### 3.1  RANDOMIZED SMOOTHING PROCESS

Unlike Cohen et al. (2019) where randomized smoothing is applied to a model, we apply smoothing directly to the data distribution $p_{\text{data}}(\mathbf{x})$. To do this, we introduce a smoothing distribution $q(\tilde{\mathbf{x}}|\mathbf{x})$ — a distribution that is symmetric and stationary (e.g. a Gaussian or Laplacian kernel) — and convolve it with $p_{\text{data}}(\mathbf{x})$ to obtain a new distribution $q(\tilde{\mathbf{x}}) \triangleq \int q(\tilde{\mathbf{x}}|\mathbf{x})p_{\text{data}}(\mathbf{x})d\mathbf{x}$. When $q(\tilde{\mathbf{x}}|\mathbf{x})$ is a normal distribution, this convolution process is equivalent to perturbing the data distribution with Gaussian

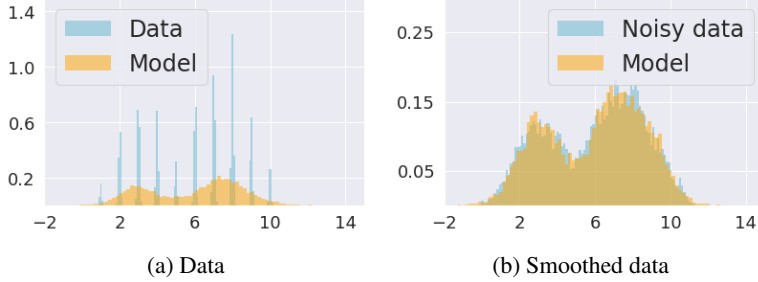

Figure 3: Visualization of a 1-d data distribution without smoothing (a) or with smoothing (b), modeled by the same mixture of logistics model.

noise, which, intuitively, will make the data distribution smoother. In the following, we formally prove that convolving a 1-d distribution $p_{\text{data}}(x)$ with a suitable noise can indeed "smooth" $p_{\text{data}}(x)$.

**Theorem 1.** *Given a continuous and bounded 1-d distribution $p_{data}(x)$ that is supported on $\mathbb{R}$, for any 1-d distribution $q(\tilde{x}|x)$ that is symmetric (i.e. $q(\tilde{x}|x) = q(x|\tilde{x})$), stationary (i.e. translation invariant) and satisfies $\lim_{x \to \infty} p_{data}(x)q(x|\tilde{x}) = 0$ for any given $\tilde{x}$, we have $Lip(q(\tilde{x})) \leq Lip(p_{data}(x))$, where $q(\tilde{x}) \triangleq \int q(\tilde{x}|x)p_{data}(x)dx$ and $Lip(\cdot)$ denotes the Lipschitz constant of the given 1-d function.*

Theorem 1 shows that convolving a 1-d data distribution $p_{\text{data}}(x)$ with a suitable noise distribution $q(\tilde{x}|x)$ (*e.g.* $\mathcal{N}(\tilde{x}|x, \sigma^2)$) can reduce the Lipschitzness (*i.e.* increase the smoothness) of $p_{\text{data}}(x)$. We provide the proof of Theorem 1 in Appendix A.

Given $p_{\text{data}}(\mathbf{x})$ with a high Lipschitz constant, we empirically verify that density estimation becomes an easier task on the smoothed distribution $q(\tilde{\mathbf{x}})$ than directly on $p_{\text{data}}(\mathbf{x})$. To see this, we visualize a 1-d example in figure 3a, where we want to model a ten-mode data distribution with a mixture of logistics model. If our model has three logistic components, there is almost no way for the model, which only has three modes, to perfectly fit this data distribution, which has ten separate modes with sharp transitions. The model, after training (see figure 3a), mistakenly assigns a much higher density to the low density regions between nearby modes. If we convolve the data distribution with $q(\tilde{x}|x) = \mathcal{N}(\tilde{x}|x, 0.5^2)$, the new distribution becomes smoother (see figure 3b) and can be captured reasonably well by the same mixture of logistics model with only three modes (see figure 3b). Comparing the same model's performance on the two density estimation tasks, we can see that the model is doing a better job at modeling the smoothed version of the data distribution than the original data distribution, which has a high Lipschitz constant.

This smoothing process can also be understood as a regularization term for the original maximum likelihood objective (on the un-smoothed data distribution), encouraging the learned model to be smooth, as formalized by the following statement:

**Proposition 1** (Informal)**.** *Assume that the symmetric and stationary smoothing distribution $q(\tilde{\mathbf{x}}|\mathbf{x})$ has small variance and negligible higher order moments, then*

$$\mathbb{E}_{p_{data}(\mathbf{x})}\mathbb{E}_{q(\tilde{\mathbf{x}}|\mathbf{x})}[\log p_\theta(\tilde{\mathbf{x}})] \approx \mathbb{E}_{p_{data}(\mathbf{x})}\left[\log p_\theta(\mathbf{x}) + \frac{\eta}{2}\sum_i \frac{\partial^2 \log p_\theta}{\partial x_i^2}\right],$$

*for some constant $\eta$.*

Proposition 1 shows that our smoothing process provides a regularization effect on the original objective $\mathbb{E}_{p_{\text{data}}(\mathbf{x})}[\log p_\theta(\mathbf{x})]$ when no noise is added, where the regularization aims to maximize $\frac{\eta}{2}\sum_i \frac{\partial^2 \log p_\theta}{\partial x_i^2}$. Since samples from $p_{\text{data}}$ should be close to a local maximum of the model, this encourages the second order gradients computed at a data point $\mathbf{x}$ to become closer to zero (if it were positive then $\mathbf{x}$ will not be a local maximum), creating a smoothing effect. This extra term is also the trace of the score function (up to a multiplicative constant) that can be found in the score matching objective (Hyvärinen, 2005), which is closely related to many denoising methods (Vincent, 2011; Hyvärinen, 2008). This regularization effect can, intuitively, increase the generalization capability of the model. In fact, it has been demonstrated empirically that training with noise can lead to improvements in network generalization (Sietsma & Dow, 1991; Bishop, 1995). Our argument is also

similar to that used in (Bishop, 1995) except that we consider a more general generative modeling case as opposed to supervised learning with squared error. We provide the formal statement and proof of Proposition 1 in Appendix A.

## 3.2 Autoregressive distribution smoothing models

Motivated by the previous 1-d example, instead of directly modeling $p_{\text{data}}(\mathbf{x})$, which can have a high Lipschitz constant, we propose to first train an autoregressive model on the smoothed version of the data distribution $q(\tilde{\mathbf{x}})$. Although the smoothing process makes the distribution easier to learn, it also introduces bias. Thus, we need an extra step to debias the learned distribution by reverting the smoothing process.

If our goal is to generate approximate samples for $p_{\text{data}}(\mathbf{x})$, when $q(\tilde{\mathbf{x}}|\mathbf{x}) = \mathcal{N}(\tilde{\mathbf{x}}|\mathbf{x}, \sigma^2 I)$ and $\sigma$ is small, we can use the gradient of $p_\theta(\tilde{\mathbf{x}})$ for denoising (Alain & Bengio, 2014). More specifically, given smoothed samples $\tilde{\mathbf{x}}$ from $p_\theta(\tilde{\mathbf{x}})$, we can "denoise" samples via:

$$\bar{\mathbf{x}} = \tilde{\mathbf{x}} + \sigma^2 \nabla_{\tilde{\mathbf{x}}} \log p_\theta(\tilde{\mathbf{x}}), \tag{2}$$

which only requires the knowledge of $p_\theta(\tilde{\mathbf{x}})$ and the ability to sample from it. However, this approach does not provide a likelihood estimate and Eq. (2) only works when $q(\tilde{\mathbf{x}}|\mathbf{x})$ is Gaussian (though alternative denoising updates for other smoothing processes could be derived under the Empirical Bayes framework (Raphan & Simoncelli, 2011)). Although Eq. (2) could provide reasonable denoising results when the smoothing distribution has a small variance, $\bar{\mathbf{x}}$ obtained in this way is only a point estimation of $\bar{\mathbf{x}} = \mathbb{E}[\mathbf{x}|\tilde{\mathbf{x}}]$ and does not capture the uncertainty of $p(\mathbf{x}|\tilde{\mathbf{x}})$.

To invert more general smoothing distributions (beyond Gaussians) and to obtain likelihood estimations, we introduce a second autoregressive model $p_\theta(\mathbf{x}|\tilde{\mathbf{x}})$. The parameterized joint density $p_\theta(\mathbf{x}, \tilde{\mathbf{x}})$ can then be computed as $p_\theta(\mathbf{x}, \tilde{\mathbf{x}}) = p_\theta(\mathbf{x}|\tilde{\mathbf{x}})p_\theta(\tilde{\mathbf{x}})$. To obtain our approximation of $p_{\text{data}}(\mathbf{x})$, we need to integrate over $\tilde{\mathbf{x}}$ on the joint distribution $p_\theta(\mathbf{x}, \tilde{\mathbf{x}})$ to obtain $p_\theta(\mathbf{x}) = \int p_\theta(\mathbf{x}, \tilde{\mathbf{x}})d\tilde{\mathbf{x}}$, which is in general intractable. However, we can easily obtain an evidence lower bound (ELBO):

$$\log p_\theta(\mathbf{x}) \geq \mathbb{E}_{q(\tilde{\mathbf{x}}|\mathbf{x})}[\log p_\theta(\tilde{\mathbf{x}})] - \mathbb{E}_{q(\tilde{\mathbf{x}}|\mathbf{x})}[\log q(\tilde{\mathbf{x}}|\mathbf{x})] + \mathbb{E}_{q(\tilde{\mathbf{x}}|\mathbf{x})}[\log p_\theta(\mathbf{x}|\tilde{\mathbf{x}})]. \tag{3}$$

Note that when $q(\tilde{\mathbf{x}}|\mathbf{x})$ is fixed, the entropy term $\mathbb{E}_{q(\tilde{\mathbf{x}}|\mathbf{x})}[\log q(\tilde{\mathbf{x}}|\mathbf{x})]$ is a constant with respect to the optimization parameters. Maximizing ELBO on $p_{\text{data}}(\mathbf{x})$ is then equivalent to maximizing:

$$J(\theta) = \mathbb{E}_{p_{\text{data}}(\mathbf{x})}\left[\mathbb{E}_{q(\tilde{\mathbf{x}}|\mathbf{x})}[\log p_\theta(\tilde{\mathbf{x}})] + \mathbb{E}_{q(\tilde{\mathbf{x}}|\mathbf{x})}[\log p_\theta(\mathbf{x}|\tilde{\mathbf{x}})]\right]. \tag{4}$$

From equation 4, we can see that optimizing the two models $p_\theta(\tilde{\mathbf{x}})$ and $p_\theta(\mathbf{x}|\tilde{\mathbf{x}})$ separately via maximum likelihood estimation is equivalent to optimizing $J(\theta)$.

## 3.3 Tradeoff in modeling

In general, there is a trade-off between the difficulty of modeling $p_\theta(\tilde{\mathbf{x}})$ and $p_\theta(\mathbf{x}|\tilde{\mathbf{x}})$. To see this, let us consider two extreme cases for the variance of $q(\tilde{\mathbf{x}}|\mathbf{x})$ — when $q(\tilde{\mathbf{x}}|\mathbf{x})$ has a zero variance and an infinite variance. When $q(\tilde{\mathbf{x}}|\mathbf{x})$ has a zero variance, $q(\tilde{\mathbf{x}}|\mathbf{x})$ is a distribution with all its probability mass at $\mathbf{x}$, meaning that no noise is added to the data distribution. In this case, modeling the smoothed distribution would be equivalent to modeling $p_{\text{data}}(\mathbf{x})$, which can be hard as discussed above. The reverse smoothing process, however, would be easy since $p_\theta(\mathbf{x}|\tilde{\mathbf{x}})$ can simply be an identity map to perfectly invert the smoothing process. In the second case when $q(\tilde{\mathbf{x}}|\mathbf{x})$ has an infinite variance, modeling $p(\tilde{\mathbf{x}})$ would be easy because all the information about the original data is lost, and $p(\tilde{\mathbf{x}})$ would be close to the smoothing distribution. Modeling $p(\mathbf{x}|\tilde{\mathbf{x}})$, on the other hand, is equivalent to directly modeling $p_{\text{data}}(\mathbf{x})$, which can be challenging.

Thus, the key here is to appropriately choose a smoothing level so that both $q(\tilde{\mathbf{x}})$ and $p(\mathbf{x}|\tilde{\mathbf{x}})$ can be approximated relatively well by existing autoregressive models. In general, the optimal variance might be hard to find. Although one can train $q(\tilde{\mathbf{x}}|\mathbf{x})$ by jointly optimizing ELBO, in practice, we find this approach often assigns a very large variance to $q(\tilde{\mathbf{x}}|\mathbf{x})$, which can trade-off sample quality for better likelihoods on high dimensional image datasets. We find empirically that a pre-specified $q(\tilde{\mathbf{x}}|\mathbf{x})$ chosen by heuristics (Saremi & Hyvarinen, 2019; Garreau et al., 2017) is able to generate much better samples than training $q(\tilde{\mathbf{x}}|\mathbf{x})$ via ELBO. In this paper, we will focus on the sample quality and leave the training of $q(\tilde{\mathbf{x}}|\mathbf{x})$ for future work.

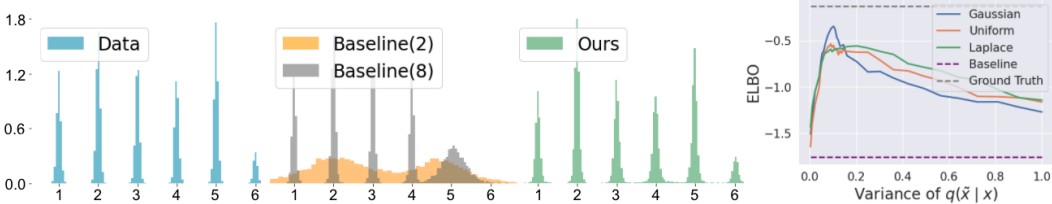

Figure 4: Density estimation on 1-d synthetic dataset. In the second figure, the digit in the parenthesis denotes the number of mixture components used in the baseline mixture of logistics model. In comparison, our model in the third figure uses only 2 mixture of logistics components for each univariate conditional distribution.

Table 1: Negative log-likelihoods on 2-d synthetic datasets (lower is better). We compare with MADE (Germain et al., 2015), RealNVP (Dinh et al., 2016), CIF-RealNVP (Cornish et al., 2019).

| Dataset | RealNVP | CIF-RealNVP | MADE (3 mixtures) | MADE (6 mixtures) | Ours (3 mixtures) |
|---------|---------|-------------|-------------------|-------------------|-------------------|
| Rings   | 2.81    | 2.81        | 3.26              | 2.81              | **2.71**          |
| Olympics| 1.80    | 1.74        | 1.27              | **0.80**          | **0.80**          |

## 4 EXPERIMENTS

In this section, we demonstrate empirically that by appropriately choosing the smoothness level of randomized smoothing, our approach is able to drastically improve the sample quality of existing autoregressive models on several synthetic and real-world datasets while retaining competitive likelihoods on synthetic datasets. We also present results on image inpainting in Appendix C.2.

### 4.1 CHOOSING THE SMOOTHING DISTRIBUTION

To help us build insights into the selection of the smoothing distribution $q(\tilde{x}|\mathbf{x})$, we first focus on a 1-d multi-modal distribution (see figure 4 leftmost panel). We use model-based methods to invert the smoothed distribution and provide analysis on "single-step denoising" in Appendix B.1. We start with the exploration of three different types of smoothing distributions – Gaussian distribution, Laplace distribution, and uniform distribution. For each type of distribution, we perform a grid search to find the optimal variance. Since our approach requires the modeling of both $p_\theta(\tilde{\mathbf{x}})$ and $p_\theta(\mathbf{x}|\tilde{\mathbf{x}})$, we stack $\tilde{\mathbf{x}}$ and $\mathbf{x}$ together, and use a MADE model (Germain et al., 2015) with a mixture of two logistic components to parameterize $p_\theta(\tilde{\mathbf{x}})$ and $p_\theta(\mathbf{x}|\tilde{\mathbf{x}})$ at the same time. For the baseline model, we train a mixture of logistics model directly on $p_{\text{data}}(\mathbf{x})$. We compare the results in the middle two panels in figure 4.

We find that although the baseline with eight logistic components has the capacity to perfectly model the multi-modal data distribution, which has six modes, the baseline model still fails to do so. We believe this can be caused by optimization or initialization issues for modeling a distribution with a high Lipschitz constant. Our method, on the other hand, demonstrates more robustness by successfully modeling the different modes in the data distribution even when using only two mixture components for both $p_\theta(\tilde{\mathbf{x}})$ and $p_\theta(\mathbf{x}|\tilde{\mathbf{x}})$.

For all the three types of smoothing distributions, we observe a reverse U-shape correlation between the variance of $q(\tilde{\mathbf{x}}|\mathbf{x})$ and ELBO values — with ELBO first increasing as the variance increases and then decreasing as the variance grows beyond a certain point. The results match our discussion on the trade-off between modeling $p_\theta(\tilde{\mathbf{x}})$ and $p_\theta(\mathbf{x}|\tilde{\mathbf{x}})$ in Section 3.3. We notice from the empirical results that Gaussian smoothing is able to obtain better ELBO than the other two distributions. Thus, we will use $q(\tilde{\mathbf{x}}|\mathbf{x}) = \mathcal{N}(\tilde{\mathbf{x}}|\mathbf{x}, \sigma^2 I)$ for the later experiments.

### 4.2 2-D SYNTHETIC DATASETS

In this section, we consider two challenging 2-d multi-modal synthetic datasets (see figure 5). We focus on model-based denoising methods and present discussion on "single-step denoising" in Appendix B.2. We use a MADE model with comparable number of total parameters for both the baseline and our approach. For the baseline, we train the MADE model directly on the data. For

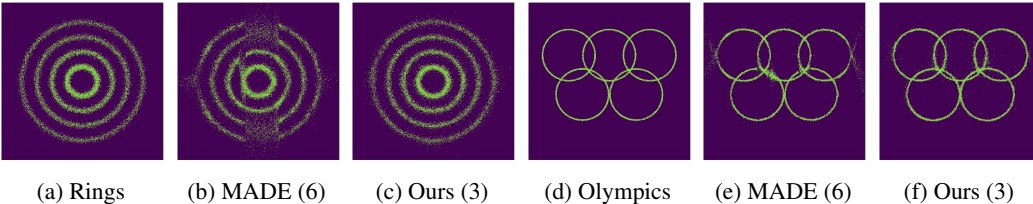

| (a) Rings | (b) MADE (6) | (c) Ours (3) | (d) Olympics | (e) MADE (6) | (f) Ours (3) |

Figure 5: Samples on 2-d synthetic datasets. We use a MADE model with comparable number of parameters for both our method and the baseline. Our model uses 3 mixture of logistics, while the baseline uses 6 (more) mixture of logistics.

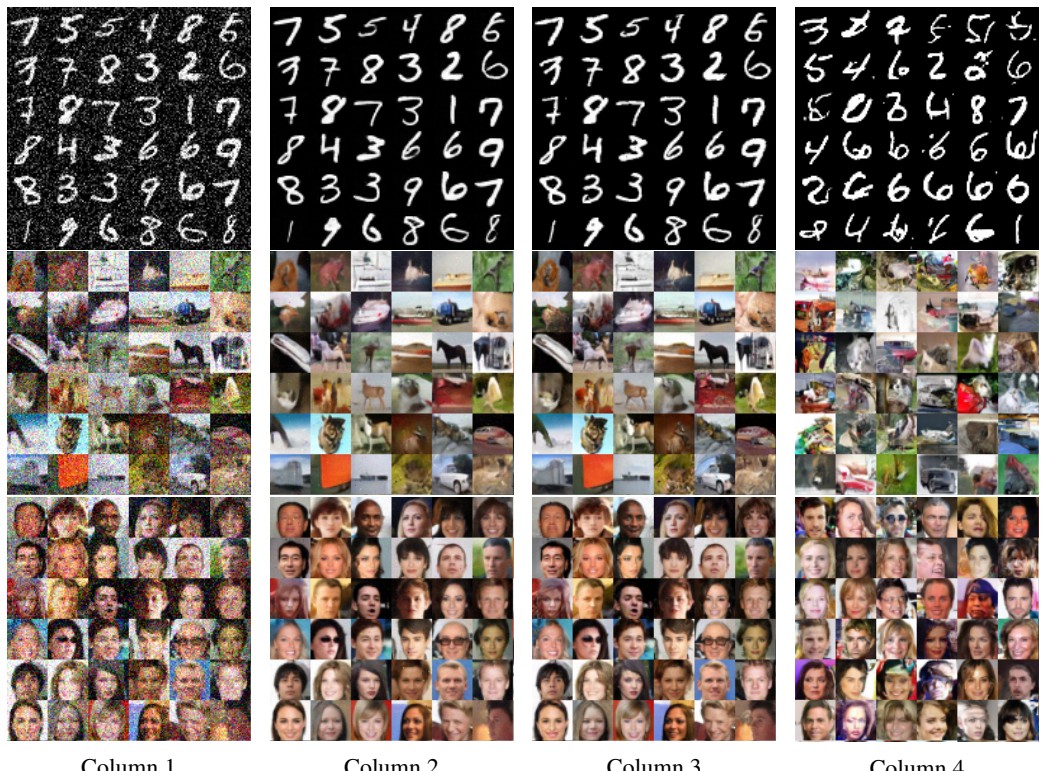

| Column 1 | Column 2 | Column 3 | Column 4 |

Figure 6: From left to right: **Column 1:** samples from $p_\theta(\tilde{\mathbf{x}})$. **Column 2:** "single-step denoising" samples from $p_\theta(\tilde{\mathbf{x}})$. **Column 3:** samples from $p_\theta(\mathbf{x}|\tilde{\mathbf{x}})$. **Column 4:** samples from the baseline PixelCNN++ model. Samples in Column 2 ("single-step denoising") contain wild pixels and are less realistic compared to samples in Column 3 (modeled by another PixelCNN++). None of the samples are conditioned on class labels.

our randomized smoothing model, we choose $q(\tilde{\mathbf{x}}|\mathbf{x}) = \mathcal{N}(\tilde{\mathbf{x}}|\mathbf{x}, 0.3^2 I)$ to be the smoothing distribution. We observe that with this randomized smoothing approach, our model is able to generate better samples than the baseline (according to a human observer) even when using less logistic components (see figure 5). We provide more analysis on the model's performance in Appendix B.2. We also provide the negative log-likelihoods in Tab. 1.

## 4.3 IMAGE EXPERIMENTS

In this section, we focus on three common image datasets, namely MNIST, CIFAR-10 (Krizhevsky et al., 2009) and CelebA (Liu et al., 2015). We select $q(\tilde{\mathbf{x}}|\mathbf{x}) = \mathcal{N}(\tilde{\mathbf{x}}|\mathbf{x}, \sigma^2 I)$ to be the smoothing distribution. We use PixelCNN++ (Salimans et al., 2017) as the model architecture for both $p_\theta(\tilde{\mathbf{x}})$ and $p_\theta(\mathbf{x}|\tilde{\mathbf{x}})$. We provide more details about settings in Appendix C.

**Image generation.** For image datasets, we select the $\sigma$ of $q(\tilde{\mathbf{x}}|\mathbf{x}) = \mathcal{N}(\tilde{\mathbf{x}}|\mathbf{x}, \sigma^2 I)$ according to analysis in (Saremi & Hyvarinen, 2019) (see Appendix C for more details). Since $q(\tilde{\mathbf{x}}|\mathbf{x})$ is a Gaussian distribution, we can apply "single-step denoising" to reverse the smoothing process for samples drawn from $p_\theta(\tilde{\mathbf{x}})$. In this case, the model $p_\theta(\mathbf{x}|\tilde{\mathbf{x}})$ is not required for sampling since the gradient of $p_\theta(\tilde{\mathbf{x}})$ can be used to denoise samples (also from $p_\theta(\tilde{\mathbf{x}})$) (see equation 2). We present smoothed samples from $p_\theta(\tilde{\mathbf{x}})$, reversed smoothing samples processed by "single-step denoising" and processed by $p_\theta(\mathbf{x}|\tilde{\mathbf{x}})$ in figure 6. For comparison, we also present samples from a PixelCNN++ with parameters comparable to the sum of total parameters of $p_\theta(\tilde{\mathbf{x}})$ and $p_\theta(\mathbf{x}|\tilde{\mathbf{x}})$. We find that by using this randomized smoothing approach, we are able to drastically improve the sample quality of PixelCNN++ (see the rightmost panel in figure 6). We note that with only $p_\theta(\tilde{\mathbf{x}})$, a PixelCNN++ optimized on the smoothed data, we already obtain more realistic samples compared to the original PixelCNN++ method. However, $p_\theta(\mathbf{x}|\tilde{\mathbf{x}})$ is needed to compute the likelihood lower bounds. We report the sample quality evaluated by Fenchel Inception Distance (FID (Heusel et al., 2017)), Kernel Inception Distance (KID (Bińkowski et al., 2018)), and Inception scores (Salimans et al., 2016) in Tab. 2. Although our method obtains better samples compared to the original PixelCNN++, our model has worse likelihoods as evaluated in BPDs. We believe this is because likelihood and sample quality are not always directly correlated as discussed in Theis et al. (2015). We also tried training the variance for $q(\tilde{\mathbf{x}}|\mathbf{x})$ by jointly optimizing ELBO. Although training the variance can produce better likelihoods, it does not generate samples with comparable quality as our method (*i.e.* choosing variance by heuristics). Thus, it is hard to conclusively determine what is the best way of choosing $q(\tilde{\mathbf{x}}|\mathbf{x})$. We provide more image samples in Appendix C.4 and nearest neighbors analysis in Appendix C.5.

| Model | Inception ↑ | FID ↓ | KID ↓ | BPD ↓ |
|---|---|---|---|---|
| PixelCNN (Oord et al., 2016b) | 4.60 | 65.93 | - | 3.14 |
| PixelIQN (Ostrovski et al., 2018) | 5.29 | 49.46 | - | - |
| PixelCNN++ (Salimans et al., 2017) | 5.30 | 54.25 | 0.046 | **2.92** |
| EBM (Du & Mordatch, 2019) | 6.02 | 40.58 | - | - |
| i-ResNet (Behrmann et al., 2019) | - | 65.01 | - | 3.45 |
| MADE (Germain et al., 2015) | - | - | - | 5.67 |
| Glow (Kingma & Dhariwal, 2018) | - | 46.90 | - | 3.35 |
| Single-step (Ours) | $7.50 \pm .08$ | 57.53 | 0.052 | - |
| Two-step (Ours) | $\mathbf{7.84} \pm .07$ | **29.83** | **0.022** | $\leq 3.53$ |

Table 2: Inception, FID and KID scores for unconditional CIFAR-10. "Single-step" samples are generated solely by $p_\theta(\tilde{\mathbf{x}})$. "Two-step" samples are generated by sampling from $p_\theta(\tilde{\mathbf{x}})$ and then denoised by $p_\theta(\mathbf{x}|\tilde{\mathbf{x}})$. Although samples from "single-step" might appear visually similar to samples from the "two-step" method, there is still a gap between their Inception, FID and KID scores.

## 5 ADDITIONAL EXPERIMENTS ON NORMALIZING FLOWS

In this section, we demonstrate empirically on 2-d synthetic datasets that randomized smoothing techniques can also be applied to improve the sample quality of normalizing flow models (Rezende & Mohamed, 2015). We focus on RealNVP (Dinh et al., 2016). We compare the RealNVP model trained with randomized smoothing, where we use $p_\theta(\mathbf{x}|\tilde{\mathbf{x}})$ (also a RealNVP) to revert the smoothing process, with a RealNVP trained with the original method but with comparable number of parameters. We observe that smoothing is able to improve sample quality on the datasets we consider (see figure 7) while also obtaining competitive likelihoods. On the checkerboard dataset, our method has negative log-likelihoods 3.64 while the original RealNVP has 3.72; on the Olympics dataset, our method has negative log-likelihoods 1.32 while the original RealNVP has 1.80. This example demonstrates that randomized smoothing techniques can also be applied to normalizing flow models.

## 6 RELATED WORK

Our approach shares some similarities with denoising autoencoders (DAE, Vincent et al. (2008)) which recovers a clean observation from a corrupted one. However, unlike DAE which has a train-

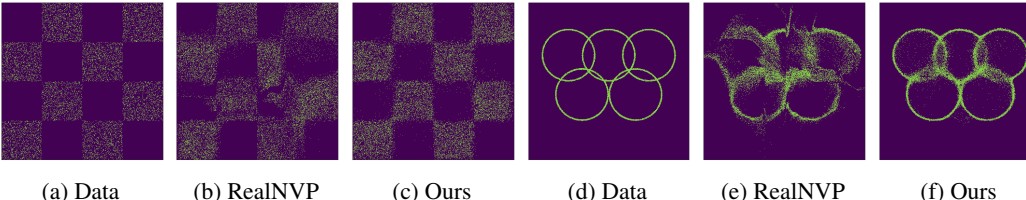

| (a) Data | (b) RealNVP | (c) Ours | (d) Data | (e) RealNVP | (f) Ours |

Figure 7: RealNVP samples on 2-d synthetic datasets. The RealNVP model trained with randomized smoothing is able to generate better samples according to human observers.

able encoder and a fixed prior distribution, our approach fixes the encoder and models the prior using an autoregressive model. Generative stochastic networks (GSN, Bengio et al. (2014)) use DAEs to train a Markov chain whose equilibrium distribution matches the data distribution. However, GSN needs to start the chain from a sample that is very close to the training distribution. Denoising diffusion model (Sohl-Dickstein et al., 2015; Ho et al., 2020) and NCSN (Song & Ermon (2019; 2020)) address the issue of GSNs by considering a sequence of distributions corresponding to data corrupted with various noise levels. By setting multiple noise levels that are close to each other, the sample from the previous level can serve as a proper initialization for the sampling process at the next level. This way, the model can start from a distribution that is easy to model and gradually move to the desired distribution. However, due to the large number of noise levels, such approaches require many steps for the chain to converge to the right data distribution.

In this paper, we instead propose to use *only one level* of smoothing by modeling each step with a powerful autoregressive model instead of deterministic autoencoders. Motivated by the success of "randomized smoothing" techniques in adversarial defense (Cohen et al., 2019), we perform randomized smoothing directly on the data distribution. Unlike denoising score matching (Vincent, 2011), a technique closely related to denoising diffusion models and NCSN, which requires the perturbed noise to be a Gaussian distribution, we are able to work with different noise distributions.

Our smoothing method is also relevant to "dequantization" approaches that are common in normalizing flow models, where the discrete data distribution is converted to a continuous one by adding continuous noise (Uria et al., 2013; Ho et al., 2019). However the added noise for "dequantization" in flows is often indistinguishable to human eyes, and the reverse "dequantization" process is often ignored. In contrast, we consider noise scales that are significantly larger and thus a denoising process is required.

Our method is also related to "quantization" approaches which reduce the number of "significant" bits that are modeled by a generative model (Kingma & Dhariwal, 2018; Menick & Kalchbrenner, 2018). For instance, Glow (Kingma & Dhariwal, 2018) only models the 5 most significant bits of an image, which improves the visual quality of samples but decreases color fidelity. SPN (Menick & Kalchbrenner, 2018) introduces another network to predict the remaining bits conditioned on the 3 most significant bits already modeled. Modeling the most significant bits can be understood as capturing a data distribution perturbed by bit-wise correlated noise, similar to modeling smoothed data in our method. Modeling the remaining bits conditioned on the most significant ones in SPN is then similar to denoising. However, unlike these quantization approaches which process an image at the "significant" bits level, we apply continuous data independent Gaussian noise to the entire image with a different motivation to smooth the data density function.

## 7 DISCUSSION

In this paper, we propose to incorporate randomized smoothing techniques into autoregressive modeling. By choosing the smoothness level appropriately, this seemingly simple approach is able to drastically improve the sample quality of existing autoregressive models on several synthetic and real-world datasets while retaining reasonable likelihoods. Our work provides insights into how recent adversarial defense techniques can be leveraged to building more robust generative models. Since we apply randomized smoothing technique directly to the target data distribution other than the model, we believe our approach is also applicable to other generative models such as variational autoencoders (VAEs) and generative adversarial networks (GANs).

## ACKNOWLEDGEMENTS

The authors would like to thank Kristy Choi for reviewing the draft of the paper. This research was supported by NSF (#1651565, #1522054, #1733686), ONR (N00014-19-1-2145), AFOSR (FA9550-19-1-0024), ARO, and Amazon AWS.

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

## A  PROOFS

**Theorem 1.** *Given a continuous and bounded 1-d distribution $p_{data}(x)$ that is supported on $\mathbb{R}$, for any 1-d distribution $q(\tilde{x}|x)$ that is symmetric (i.e. $q(\tilde{x}|x) = q(x|\tilde{x})$), stationary (i.e. translation invariant) and satisfies $\lim_{x\to\infty} p_{data}(x)q(x|\tilde{x}) = 0$ for any given $\tilde{x}$, we have $Lip(q(\tilde{x})) \leq Lip(p_{data}(x))$, where $q(\tilde{x}) \triangleq \int q(\tilde{x}|x)p_{data}(x)dx$ and $Lip(\cdot)$ denotes the Lipschitz constant of the given 1-d function.*

*Proof.* First, we have that:

$$|\nabla_{\boldsymbol{x}} p(\boldsymbol{x})| = |p(\boldsymbol{x})\nabla_{\boldsymbol{x}} \log p(\boldsymbol{x})| \leq Lip(p) \tag{5}$$

and if we assume symmetry, i.e. $q(\boldsymbol{x}|\tilde{\boldsymbol{x}}) = q(\tilde{\boldsymbol{x}}|\boldsymbol{x})$ then by integration by parts we have:

$$\nabla_{\tilde{\boldsymbol{x}}} q(\tilde{\boldsymbol{x}}) = \mathbb{E}_{p(\boldsymbol{x})}[\nabla_{\tilde{\boldsymbol{x}}} q(\tilde{\boldsymbol{x}}|\boldsymbol{x})] = \mathbb{E}_{p(\boldsymbol{x})}[\nabla_{\boldsymbol{x}} q(\boldsymbol{x}|\tilde{\boldsymbol{x}})] = -\mathbb{E}_{p(\boldsymbol{x})}[q(\boldsymbol{x}|\tilde{\boldsymbol{x}})\nabla_{\boldsymbol{x}} \log p(\boldsymbol{x})] \tag{6}$$

Therefore,

$$Lip(q) = \max_{\tilde{\boldsymbol{x}}}|-\mathbb{E}_{p(\boldsymbol{x})}[q(\boldsymbol{x}|\tilde{\boldsymbol{x}})\nabla_{\boldsymbol{x}} \log p(\boldsymbol{x})]| = \max_{\tilde{\boldsymbol{x}}}|\sum_{\boldsymbol{x}} q(\boldsymbol{x}|\tilde{\boldsymbol{x}})p(\boldsymbol{x})\nabla_{\boldsymbol{x}} \log p(\boldsymbol{x})| \tag{7}$$

$$\leq \max_{\tilde{\boldsymbol{x}}}|\sum_{\boldsymbol{x}} q(\boldsymbol{x}|\tilde{\boldsymbol{x}})Lip(p)| = Lip(p) \max_{\tilde{\boldsymbol{x}}}|\sum_{\boldsymbol{x}} q(\boldsymbol{x}|\tilde{\boldsymbol{x}})| = Lip(p) \tag{8}$$

which proves the result. $\qquad\square$

**Proposition 1** (Formal). *Given a D-dimensional data distribution $p_{data}(\mathbf{x})$ and model distribution $p_\theta(\mathbf{x})$, assume that the smoothing distribution $q(\tilde{\mathbf{x}}|\mathbf{x})$ satisfies:*

- *$\log p_\theta$ is infinitely differentiable on the support of $p_\theta(\mathbf{x})$*

- *$q(\tilde{\mathbf{x}}|\mathbf{x})$ is symmetric (i.e. $q(\tilde{\mathbf{x}}|\mathbf{x}) = q(\mathbf{x}|\tilde{\mathbf{x}})$)*

- *$q(\tilde{\mathbf{x}}|\mathbf{x})$ is stationary (i.e. translation invariant)*

- *$q(\tilde{\mathbf{x}}|\mathbf{x})$ is bounded and fully supported on $\mathbb{R}^D$*

- *$q(\tilde{\mathbf{x}}|\mathbf{x})$ is element-wise independent*

- *$\mathbb{E}_{q(\tilde{\mathbf{x}}|\mathbf{x})}[(\tilde{\mathbf{x}} - \mathbf{x})^2]$ is bounded, and $\mathbb{E}_{q(\tilde{\mathbf{x}}|\mathbf{x})}[(\tilde{x}_i - x_i)^2] = \eta$ at each dimension $i$.*

*Denote $\epsilon = \tilde{\mathbf{x}} - \mathbf{x}$, then*

$$\mathbb{E}_{p_{data}(\mathbf{x})}\mathbb{E}_{q(\tilde{\mathbf{x}}|\mathbf{x})}[\log p_\theta(\tilde{\mathbf{x}})] = \mathbb{E}_{p_{data}(\mathbf{x})}\left[\log p_\theta(\mathbf{x}) + \frac{\eta}{2}\sum_i \frac{\partial^2 \log p_\theta}{\partial x_i^2}\right] + \int\int o(\epsilon^2)p_{\text{data}}(\mathbf{x})p(\epsilon)\mathrm{d}\mathbf{x}\mathrm{d}\epsilon,$$

*where $o(\epsilon^2) : \mathbb{R}^D \to \mathbb{R}$ is a function of $\epsilon$ such that $\lim_{\epsilon\to 0}\frac{o(\epsilon^2)}{\epsilon^2} = 0$. Thus when $\int\int o(\epsilon^2)p_{\text{data}}(\mathbf{x})p(\epsilon)\mathrm{d}\mathbf{x}\mathrm{d}\epsilon \to 0$, we have*

$$\mathbb{E}_{p_{data}(\mathbf{x})}\mathbb{E}_{q(\tilde{\mathbf{x}}|\mathbf{x})}[\log p_\theta(\tilde{\mathbf{x}})] \to \mathbb{E}_{p_{data}(\mathbf{x})}\left[\log p_\theta(\mathbf{x}) + \frac{\eta}{2}\sum_i \frac{\partial^2 \log p_\theta}{\partial x_i^2}\right]. \tag{2}$$

*Proof.* To see this, we first note that the new training objective for the smoothed data distribution is $\mathbb{E}_{p_{\text{data}}(\mathbf{x})}\mathbb{E}_{q(\tilde{\mathbf{x}}|\mathbf{x})}[\log p_\theta(\tilde{\mathbf{x}})]$. Let $\epsilon = \tilde{\mathbf{x}} - \mathbf{x}$, because of the assumptions we have, the PDF function $q(\tilde{\mathbf{x}}|\mathbf{x})$ can be reparameterized as $p(\epsilon)$ which satisfies: $p$ is bounded and fully supported on $\mathbb{R}^D$; $p$ is element-wise independent and $\mathbb{E}_{p(\epsilon)}[\epsilon_i^2] = \mathbb{E}_{p(\epsilon_i)}[\epsilon_i^2] = \eta$ at each dimension $i$ ($i = 1, ..., D$). Then we have

$$\mathbb{E}_{p_{\text{data}}(\mathbf{x})}\mathbb{E}_{q(\tilde{\mathbf{x}}|\mathbf{x})}[\log p_\theta(\tilde{\mathbf{x}})] = \int\int \log p_\theta(\mathbf{x} + \epsilon)p_{\text{data}}(\mathbf{x})p(\epsilon)\mathrm{d}\mathbf{x}\mathrm{d}\epsilon, \tag{9}$$

Using Taylor expansion, we have:

$$\log p_\theta(\mathbf{x} + \epsilon) = \log p_\theta(\mathbf{x}) + \sum_i \epsilon_i \frac{\partial \log p_\theta}{\partial x_i} + \frac{1}{2}\sum_{i,j} \epsilon_i\epsilon_j \frac{\partial^2 \log p_\theta}{\partial x_i \partial x_j} + o(\epsilon^2).$$

Since $\epsilon$ is independent of $\mathbf{x}$ and

$$\int_{-\infty}^{\infty} \epsilon_i \mathrm{d}\epsilon_i = 0, \quad \int_{-\infty}^{\infty}\int_{-\infty}^{\infty} \epsilon_i\epsilon_j \mathrm{d}\epsilon_i \mathrm{d}\epsilon_j = \delta_{i,j}\eta,$$

where $\delta_{i,j}$ is the Kronecker delta function, the right hand side of Equation 9 becomes

$$\mathbb{E}_{p_{\mathrm{data}}(\mathbf{x})}[\log p_\theta(\mathbf{x})] + \int\int \left( \frac{1}{2}\sum_i \epsilon_i^2 \frac{\partial^2 \log p_\theta}{\partial x_i^2} + o(\epsilon^2) \right) p_{\mathrm{data}}(\mathbf{x})p(\epsilon)\mathrm{d}\mathbf{x}\mathrm{d}\epsilon \tag{10}$$

$$= \mathbb{E}_{p_{\mathrm{data}}(\mathbf{x})}\left[ \log p_\theta(\mathbf{x}) + \frac{\eta}{2}\sum_i \frac{\partial^2 \log p_\theta}{\partial x_i^2} \right] + \int\int o(\epsilon^2)p_{\mathrm{data}}(\mathbf{x})p(\epsilon)\mathrm{d}\mathbf{x}\mathrm{d}\epsilon. \tag{11}$$

When

$$\int\int o(\epsilon^2)p_{\mathrm{data}}(\mathbf{x})p(\epsilon)\mathrm{d}\mathbf{x}\mathrm{d}\epsilon \to 0, \tag{12}$$

we have

$$\mathbb{E}_{p_{\mathrm{data}}(\mathbf{x})}\mathbb{E}_{q(\tilde{\mathbf{x}}|\mathbf{x})}[\log p_\theta(\tilde{\mathbf{x}})] \to \mathbb{E}_{p_{\mathrm{data}}(\mathbf{x})}\left[ \log p_\theta(\mathbf{x}) + \frac{\eta}{2}\sum_i \frac{\partial^2 \log p_\theta}{\partial x_i^2} \right]. \tag{13}$$

where the second term on the right hand side serves as a regularization for the original objective $\mathbb{E}_{p_{\mathrm{data}}(\mathbf{x})}[\log p_\theta(\mathbf{x})]$. $\qquad\square$

## B  DENOISING EXPERIMENTS

### B.1  ANALYSIS ON 1-D DENOISING

To provide more insights into denoising, we first study "single-step denoising" (see equation 2) on a 1-d dataset. We choose the data distribution to be a two mixture of Gaussian distribution $0.5\mathcal{N}(-0.3, 0.1^2) + 0.5\mathcal{N}(0.3, 0.1^2)$ and the smoothing distribution to be $q(\tilde{x}|x) = \mathcal{N}(\tilde{x}|x, 0.3^2)$ (see figure 8a). Since the convolution of two Gaussian distributions is also a Gaussian distribution, the smoothed data is a mixture of Gaussian distribution given by $0.5\mathcal{N}(-0.3, 0.1^2 + 0.3^2) + 0.5\mathcal{N}(0.3, 0.1^2 + 0.3^2)$. The ground truth of $\nabla_{\tilde{x}}\log p(\tilde{x})$ can then be calculated in closed form. Thus, given the smoothed data $\tilde{x}$, we can calculate the ground truth $\nabla_{\tilde{x}}\log p(\tilde{x})$ in equation 2 and obtain $\bar{x}$ using "single-step denoising". We visualize the denoising results in figure 8b. We find that the low density region between the two modes in $p_{\mathrm{data}}(x)$ are not modeled properly in figure 8b. However, this is very expected since "single-step denosing" uses $\bar{x} = \mathbb{E}[x|\tilde{x}]$ as the substitute for the denoised result. When the smoothing distribution has a large variance (like in figure 8a where the smoothed data has merged into a one mode distribution), datapoints like $\tilde{x}_0$ in the middle low density region of $p_{\mathrm{data}}(x)$ can have high density in the smoothed distribution. Since $\tilde{x}_0$, as well as other points in the middle low density region of $p_{\mathrm{data}}(x)$, can come from both modes of $p_{\mathrm{data}}(x)$ with high probability before the smoothing process (see figure 11a), the denoised $\bar{x} = \mathbb{E}[x|\tilde{x} = \tilde{x}_0]$ can still be located in the middle low density region (see figure 8b). Since a large proportion of the smoothed data is located in the middle low density region of $p_{\mathrm{data}}(x)$, we would expect certain proportion of the density to remain in the low density region after "single-step denoising" just as shown in figure 8b. However, when the smoothing distribution has a smaller variance, "single-step denoising" can achieve much better denoising results (see figure 9, where we use $q(\tilde{x}|x) = \mathcal{N}(\tilde{x}|x, 0.1^2)$). Although denoising can be easier when the smoothing distribution has a smaller variance, modeling the smoothed distribution could be harder as we discussed before.

In general, the right denoising results should be samples coming from $p(x|\tilde{x})$, which is the reason why samples from $p_\theta(x|\tilde{x})$ (i.e. introducing the model $p_\theta(x|\tilde{x})$) is more ideal than using $\mathbb{E}_\theta[x|\tilde{x}]$ as a denoising substitute (i.e. "single-step denoising"). In general, the capacity of the denoising model $p_\theta(x|\tilde{x})$ also matters in terms of denoising results. Let us again consider the datapoint $\tilde{x}_0$ shown in figure 8a. If the invert smoothing model $p_\theta(x|\tilde{x})$ is a one mode logistic distribution, due to the mode covering property of maximum likelihood estimation, given the smoothed observation $\tilde{x}_0$, the best the model can do is to center its only mode at $\tilde{x}_0$ for approximating $p(x|\tilde{x} = \tilde{x}_0)$ (see figure 10c). Thus, like $\tilde{x}_0$, the smoothed datapoints at the low density region between the two modes of $p_{\mathrm{data}}(x)$

are still likely to remain between the two modes after denoising (see figure 10b). To solve this issue, we can increase the capacity of $p_\theta(x|\tilde{x})$ by making it a two mixture of logistics. In this case, the distribution $p_\theta(x|\tilde{x} = \tilde{x}_0)$ can be captured in a better way (see figure 11c and figure 11a). After the invert smoothing process, like $\tilde{x}_0$, most smoothed datapoints in the low density can be mapped to one of the two high density modes (see figure 11b), resulting in much better denoising effects.

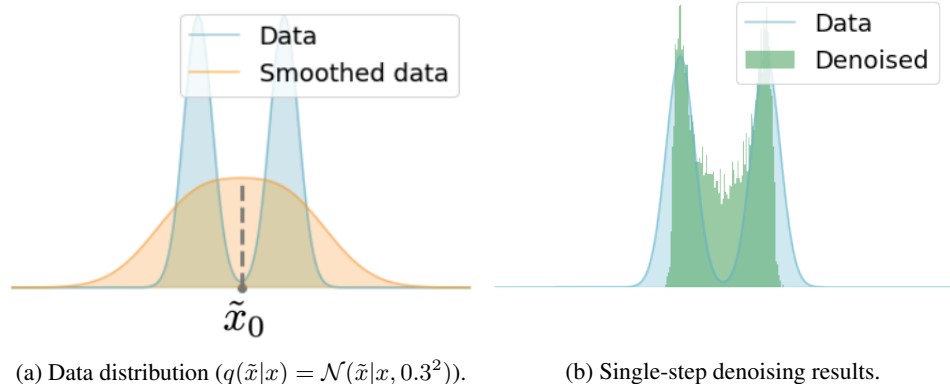

(a) Data distribution $(q(\tilde{x}|x) = \mathcal{N}(\tilde{x}|x, 0.3^2))$.  (b) Single-step denoising results.

Figure 8: 1-d single-step (gradient based) denoising.

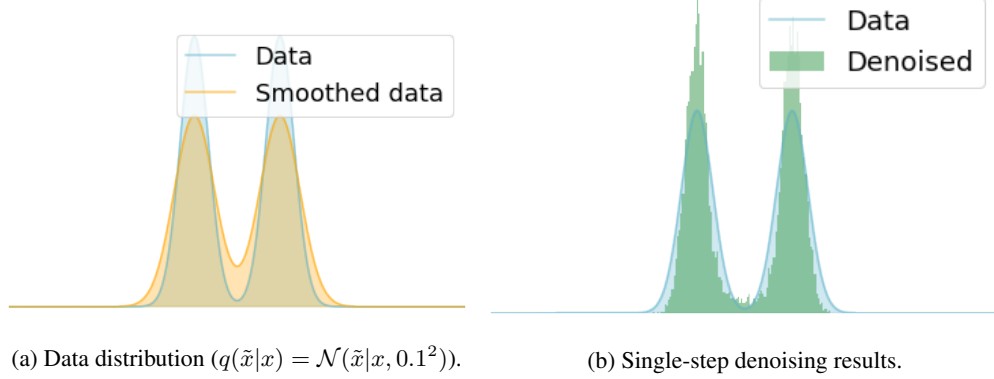

(a) Data distribution $(q(\tilde{x}|x) = \mathcal{N}(\tilde{x}|x, 0.1^2))$.  (b) Single-step denoising results.

Figure 9: 1-d single-step (gradient based) denoising.

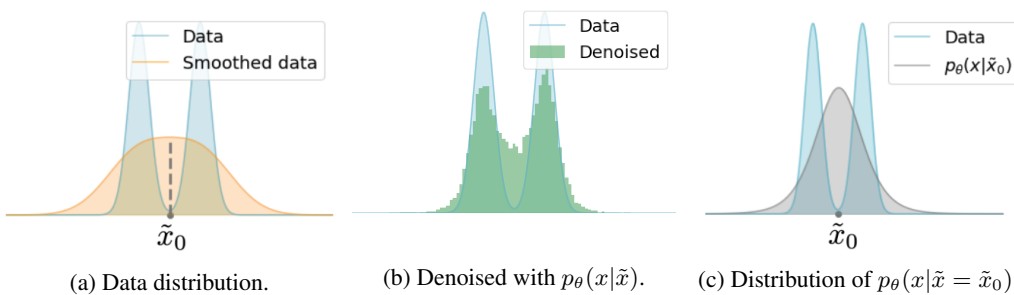

(a) Data distribution.  (b) Denoised with $p_\theta(x|\tilde{x})$.  (c) Distribution of $p_\theta(x|\tilde{x} = \tilde{x}_0)$.

Figure 10: Denoising with $p_\theta(x|\tilde{x})$, which is modeled by one mixture of logistics.

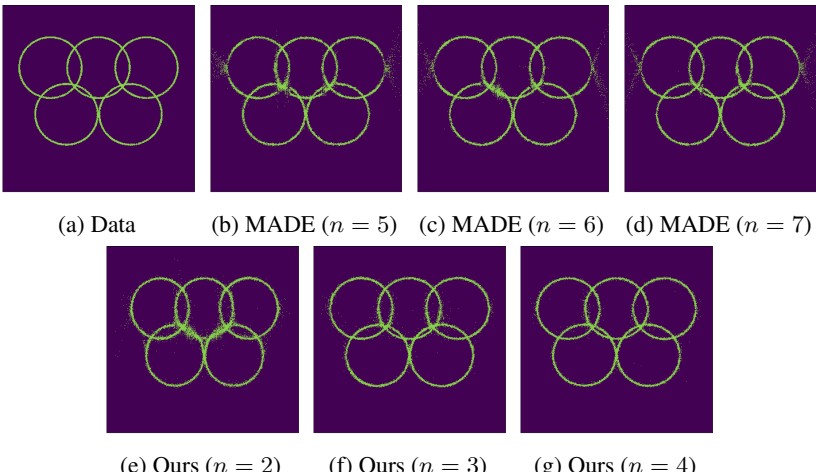

(a) Data    (b) MADE ($n = 5$)   (c) MADE ($n = 6$)   (d) MADE ($n = 7$)

(e) Ours ($n = 2$)    (f) Ours ($n = 3$)    (g) Ours ($n = 4$)

Figure 12: Samples on 2-d synthetic datasets. We use a MADE model with comparable number of parameters for both our method and the baseline. The models have $n$ mixture of logistics for each dimension. Our method is able to obtain reasonable samples when using fewer mixture components, while the baseline still has trouble modeling the two sides of the rings when $n = 7$.

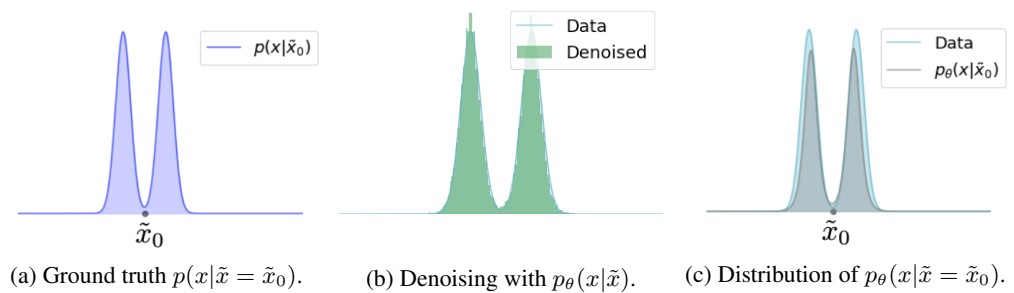

(a) Ground truth $p(x|\tilde{x} = \tilde{x}_0)$.    (b) Denoising with $p_\theta(x|\tilde{x})$.    (c) Distribution of $p_\theta(x|\tilde{x} = \tilde{x}_0)$.

Figure 11: Denoising with $p_\theta(x|\tilde{x})$, which is modeled by two mixtures of logistics.

## B.2  ANALYSIS ON 2-D DENOISING

On the 2-d Olympics dataset in section 4.2, we find that the intersections between rings can be poorly modeled with the proposed smoothing approach when only two mixture of logistics are used (see figure 12e). We believe this can be caused if the denoising model is not flexible enough to capture the distribution $p(\mathbf{x}|\tilde{\mathbf{x}})$. More specifically, we note that the ground truth distribution for $p(\mathbf{x}|\tilde{\mathbf{x}})$ at the intersections of the rings is a highly complicated distribution and can be hard to capture using our model which only has two mixtures of logistics for each dimension. If we increase the flexibility of $p_\theta(\mathbf{x}|\tilde{\mathbf{x}})$ by using three or four mixtures of logistics components (note that we still use fewer mixture components than the MADE baseline and we use comparable number of parameters), the intersection of the rings can be modeled in an improved way (see figure 12).

We also provide "single-step denoising" results for the experiments in Section 4.2 (see figure 13), where we use the same smoothing distribution, and the MADE model with three mixture components as used in section 4.2. We note that "single-step denoising" results are not very good, which is also expected. As discussed in section B.1, when the smoothing distribution has a relatively large variance, $\mathbb{E}_\theta[\mathbf{x}|\tilde{\mathbf{x}}]$ is not a good approximation for the denoised result, and we want the denoised sample to come from the distribution $p_\theta(\mathbf{x}|\tilde{\mathbf{x}})$, in which case introducing a denoising model $p_\theta(\mathbf{x}|\tilde{\mathbf{x}})$ could be a better option. Although we could select $q(\tilde{\mathbf{x}}|\mathbf{x})$ to have a smaller variance so that "single-step denoing" could work reasonably well, but modeling $p(\tilde{\mathbf{x}})$ in this case could be more challenging.

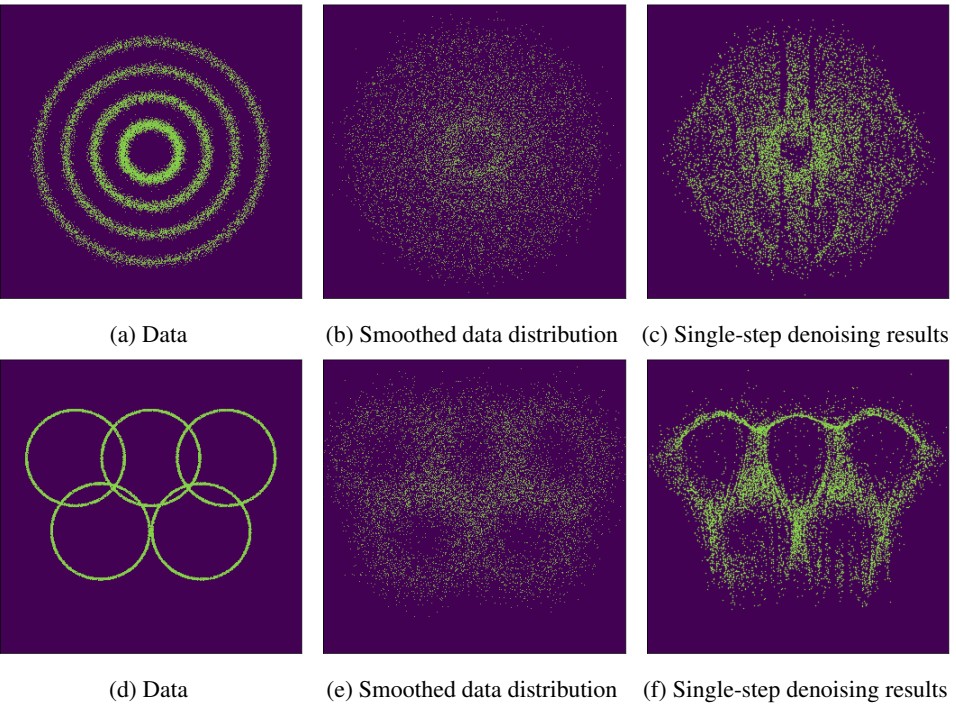

(a) Data        (b) Smoothed data distribution     (c) Single-step denoising results

(d) Data        (e) Smoothed data distribution     (f) Single-step denoising results

Figure 13: Single-step denoising results on 2-d synthetic datasets. We use the same MADE model with three mixture components and the same smoothing distribution as mentioned in Section 4.2.

## C    IMAGE EXPERIMENTS

### C.1    SETTINGS

For the image experiments, we first rescale images to $[-1, 1]$ and then perturb the images with $q(\tilde{\mathbf{x}}|\mathbf{x}) = \mathcal{N}(\tilde{\mathbf{x}}|\mathbf{x}, \sigma^2 I)$. We use $\sigma = 0.5$ for MNIST and $\sigma = 0.3$ for both CIFAR-10 and CelebA. The selection of $\sigma$ is mainly based on analysis in (Saremi & Hyvarinen, 2019). More specifically, given an image, we consider the median value of the Euclidean distance between two data points in a dataset, and then divide it by $2\sqrt{D}$, where $D$ is the dimension of the data. This provides us with a way of selecting the variance of $q(\tilde{\mathbf{x}}|\mathbf{x})$, when $q(\tilde{\mathbf{x}}|\mathbf{x})$ is a Gaussian distribution. We find this selection of variance able to generate reasonably well samples in practice. We train all the models with Adam optimizer with learning rate 0.0002. To model $p_\theta(\mathbf{x}|\tilde{\mathbf{x}})$, we stack $\tilde{\mathbf{x}}$ and $\mathbf{x}$ together at the second dimension to obtain $\hat{\mathbf{x}} = [\tilde{\mathbf{x}}, \mathbf{x}]$, which ensures that $\tilde{\mathbf{x}}$ comes before $\mathbf{x}$ in the pixel ordering. For instance, this stacking would provide an image $\hat{\mathbf{x}}$ with size $1 \times (2 \times 28) \times 28$ on a MNIST image, and an image with size $3 \times (2 \times 32) \times 32$ on a CIFAR-10 image. Since PixelCNN++ consists of convolutional layers, we can directly feed $\hat{\mathbf{x}}$ into the default architecture without modifying the model architecture. As the latter pixels of the input only depend on the previous pixels in an autoregressive model and $\tilde{\mathbf{x}}$ comes before $\mathbf{x}$, we can parameterize $p_\theta(\mathbf{x}|\tilde{\mathbf{x}})$ by computing the likelihoods only on $\mathbf{x}$ using the outputs from the autoregressive model.

### C.2    IMAGE INPAINTING

Since both $p_\theta(\tilde{\mathbf{x}})$ and $p_\theta(\mathbf{x}|\tilde{\mathbf{x}})$ are parameterized by an autoregressive model, we can also perform image inpainting using our method. We present the inpainting results on CIFAR-10 in figure 14a and CelebA in figure 14b, where the bottom half of the input image is being inpainted.

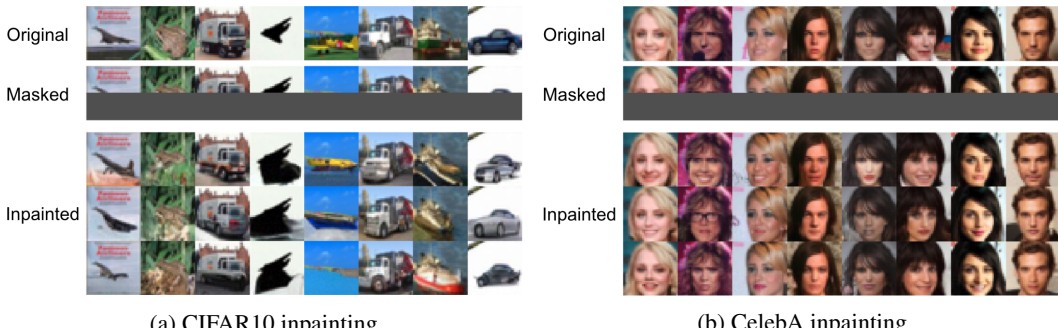

(a) CIFAR10 inpainting  (b) CelebA inpainting

Figure 14: Inpainting results from our two-step method. The bottom half of the images are masked for inpainting.

### C.3 IMAGE DENOISING

We notice that the reverse smoothing process can also be understood as a denoising process. Besides the "single-step denoising" approach shown above, we can also apply $p_\theta(\mathbf{x}|\tilde{\mathbf{x}})$ to denoise images. To visualize the denoising performance, we sample $\mathbf{x}_\text{test}$ from the test set and perturb $\mathbf{x}_\text{test}$ with $q(\tilde{\mathbf{x}}|\mathbf{x})$ to obtain a noisy sample $\tilde{\mathbf{x}}_\text{test}$. We feed $\tilde{\mathbf{x}}_\text{test}$ into $p_\theta(\mathbf{x}|\tilde{\mathbf{x}} = \tilde{\mathbf{x}}_\text{test})$ and draw samples from the model. We visualize the results in figure 15. As we can see, the model exhibits reasonable denoising results, which shows that the autoregressive model is capable of learning the data distribution when conditioned on the smoothed data.

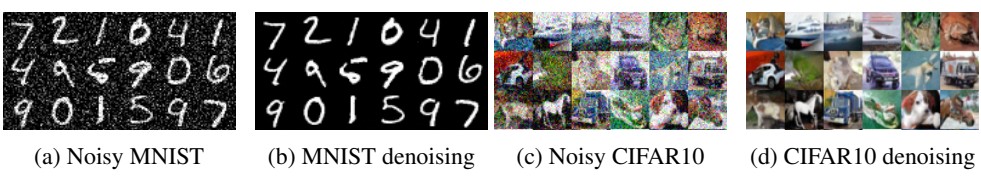

(a) Noisy MNIST  (b) MNIST denoising  (c) Noisy CIFAR10  (d) CIFAR10 denoising

Figure 15: Denoising with $p_\theta(\mathbf{x}|\tilde{\mathbf{x}})$

## C.4 MORE SAMPLES

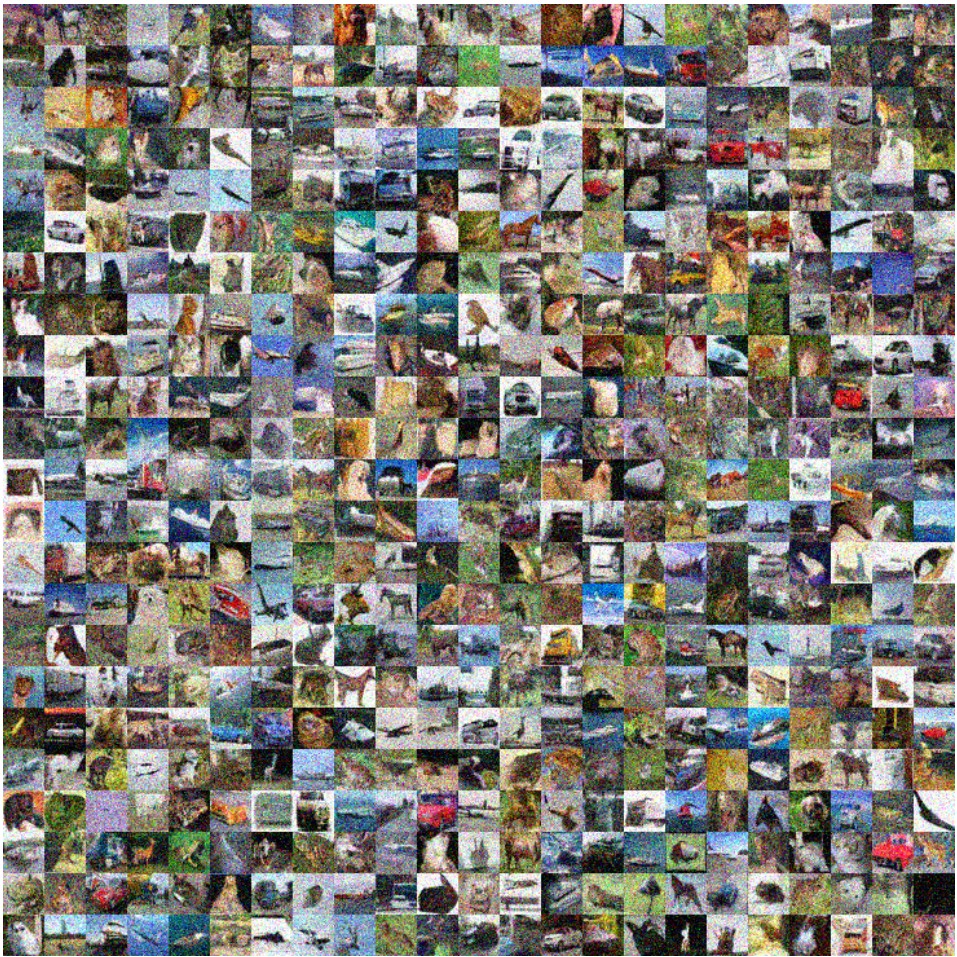

Figure 16: CIFAR-10 samples from $p_\theta(\tilde{\mathbf{x}})$ (unconditioned on class labels).

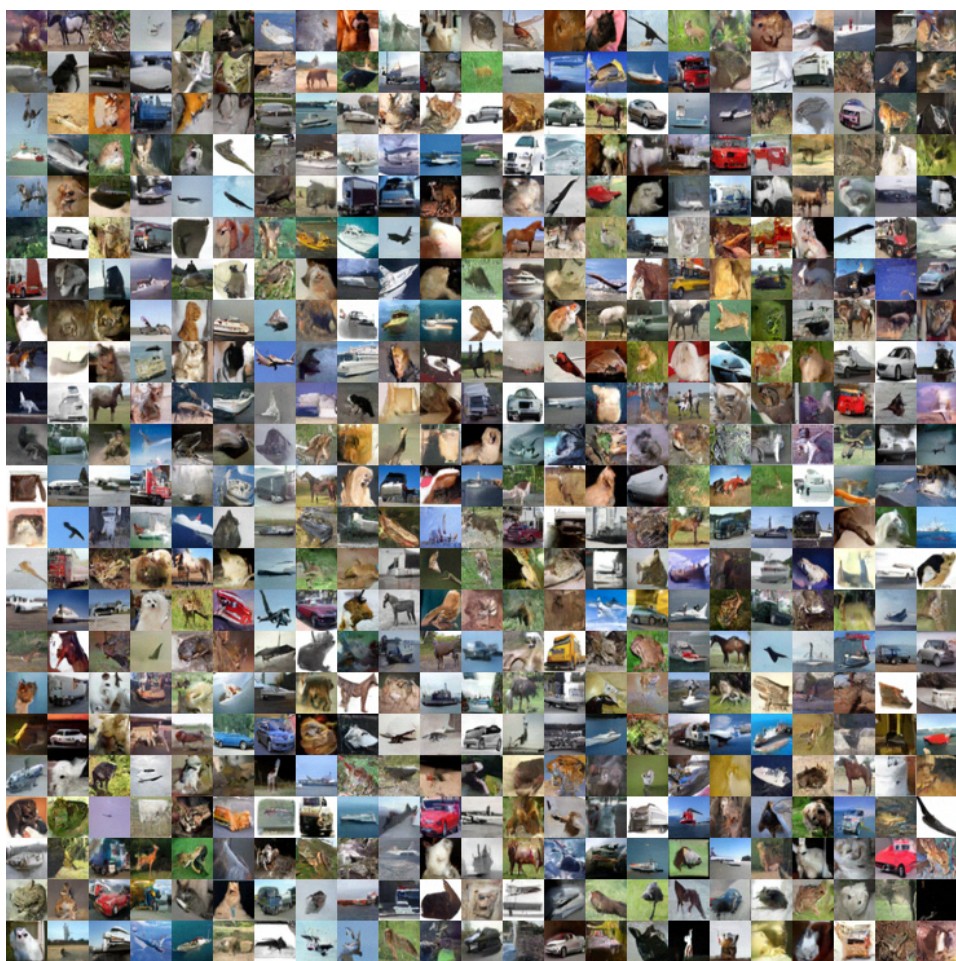

Figure 17: CIFAR-10 samples from $p_\theta(\mathbf{x}|\tilde{\mathbf{x}})$ (unconditioned on class labels).

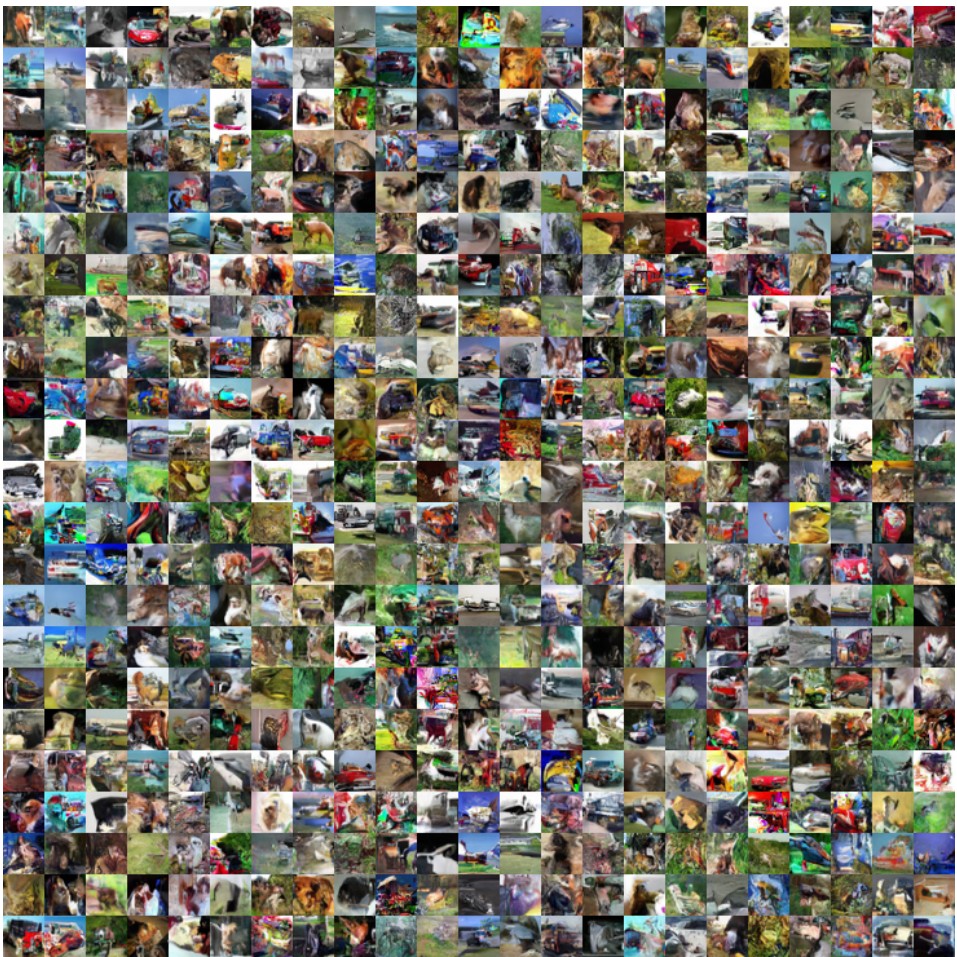

Figure 18: CIFAR-10 samples from the original PixelCNN++ method (unconditioned on class labels).

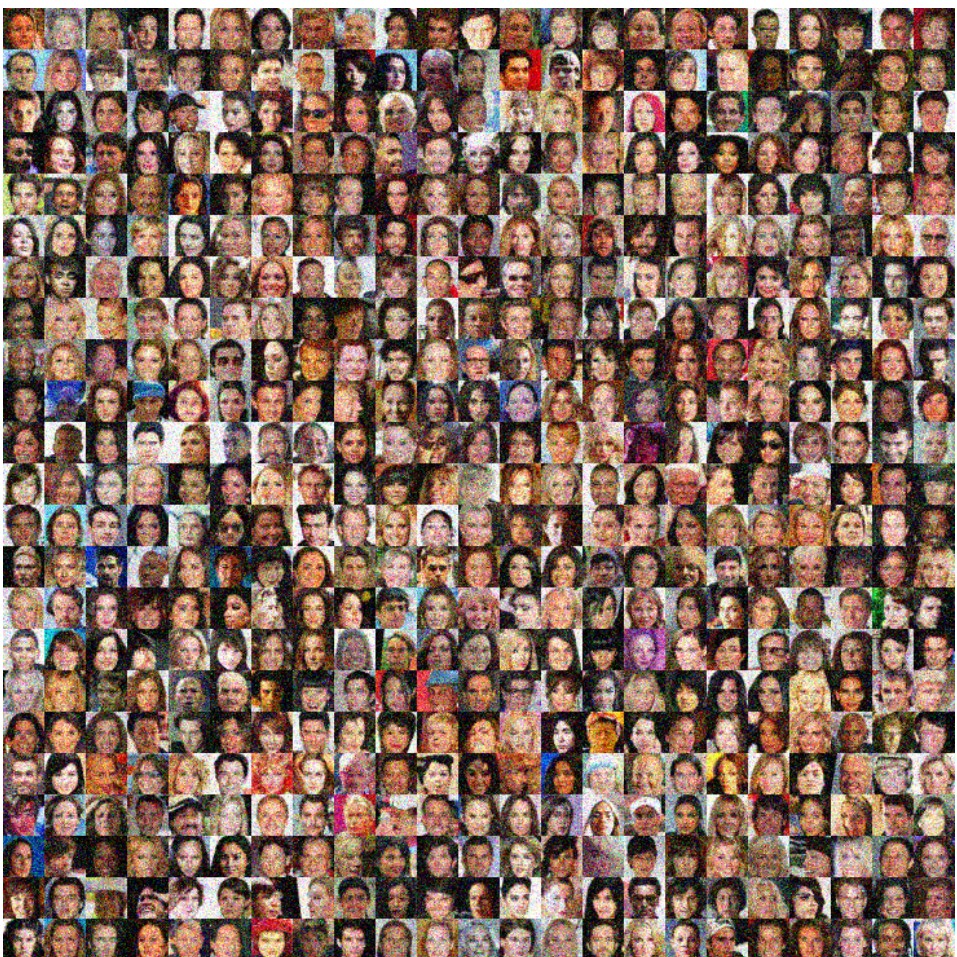

Figure 19: CelebA samples from $p_\theta(\tilde{\mathbf{x}})$ (unconditioned on class labels).

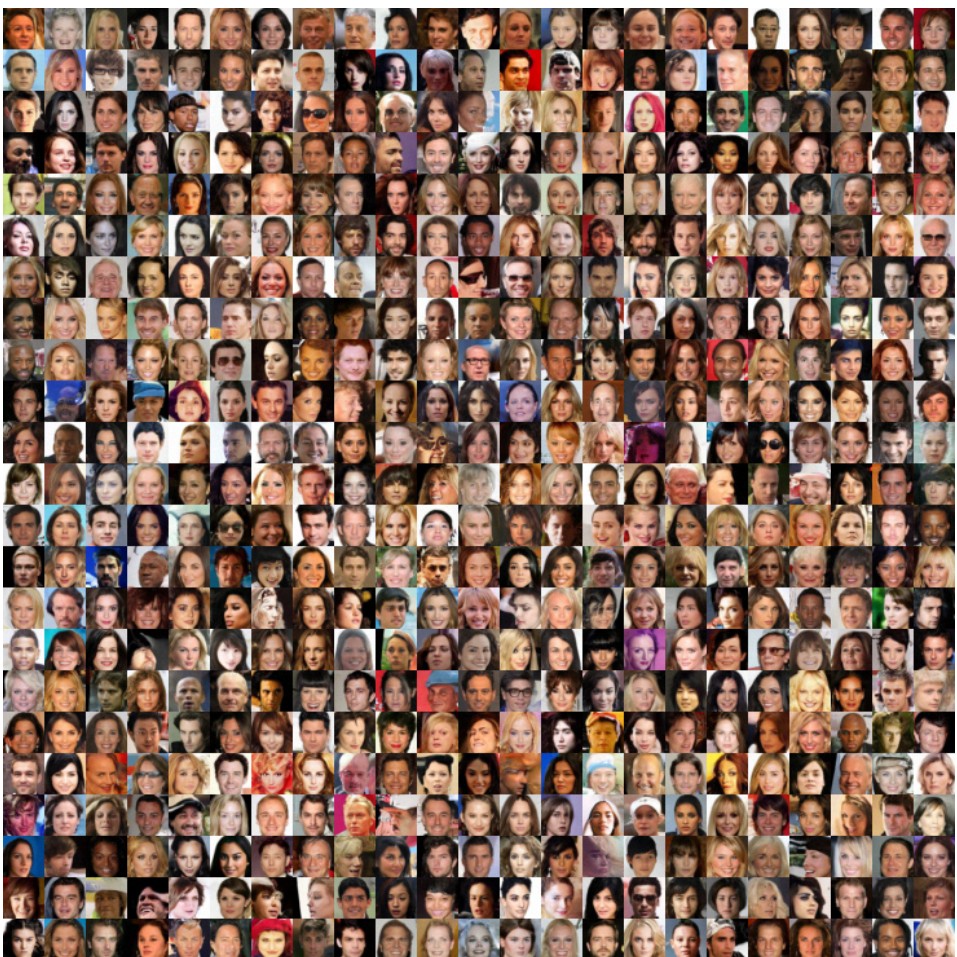

Figure 20: CelebA samples from $p_\theta(\mathbf{x}|\tilde{\mathbf{x}})$ (unconditioned on class labels).

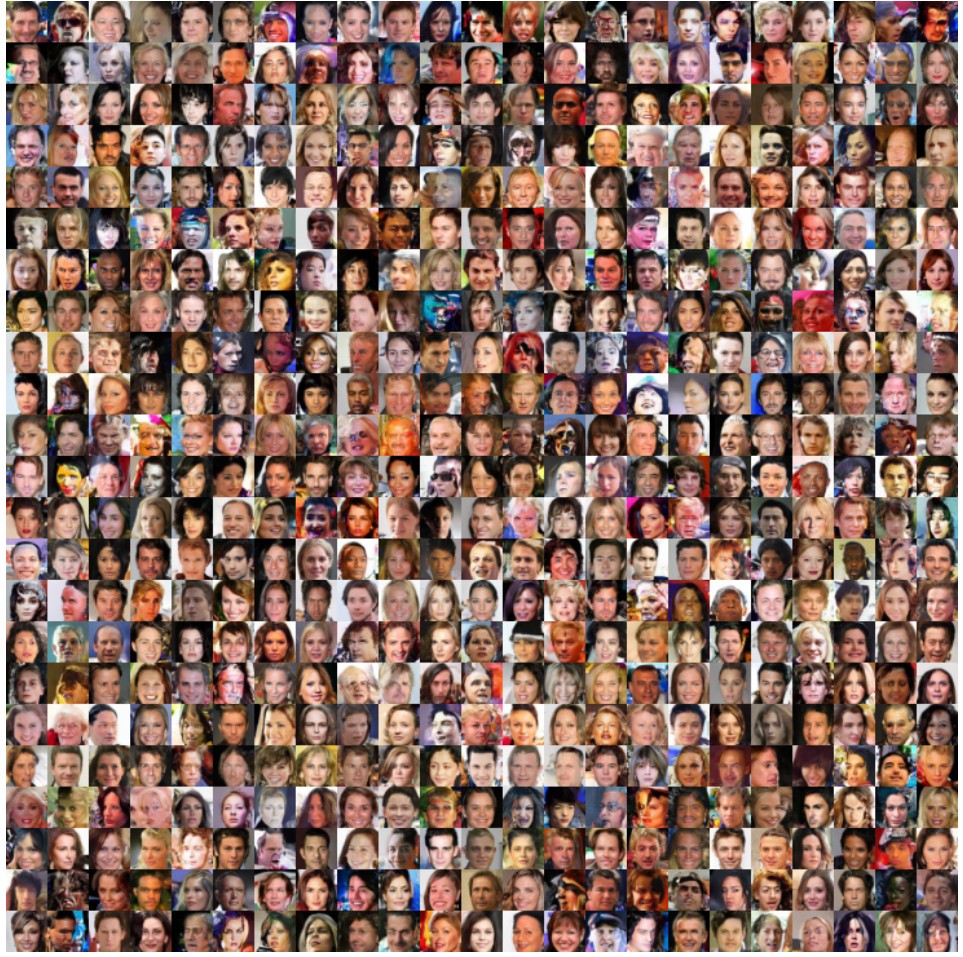

Figure 21: CelebA samples from the original PixelCNN++ method (unconditioned on class labels).

## C.5 NEAREST NEIGHBORS

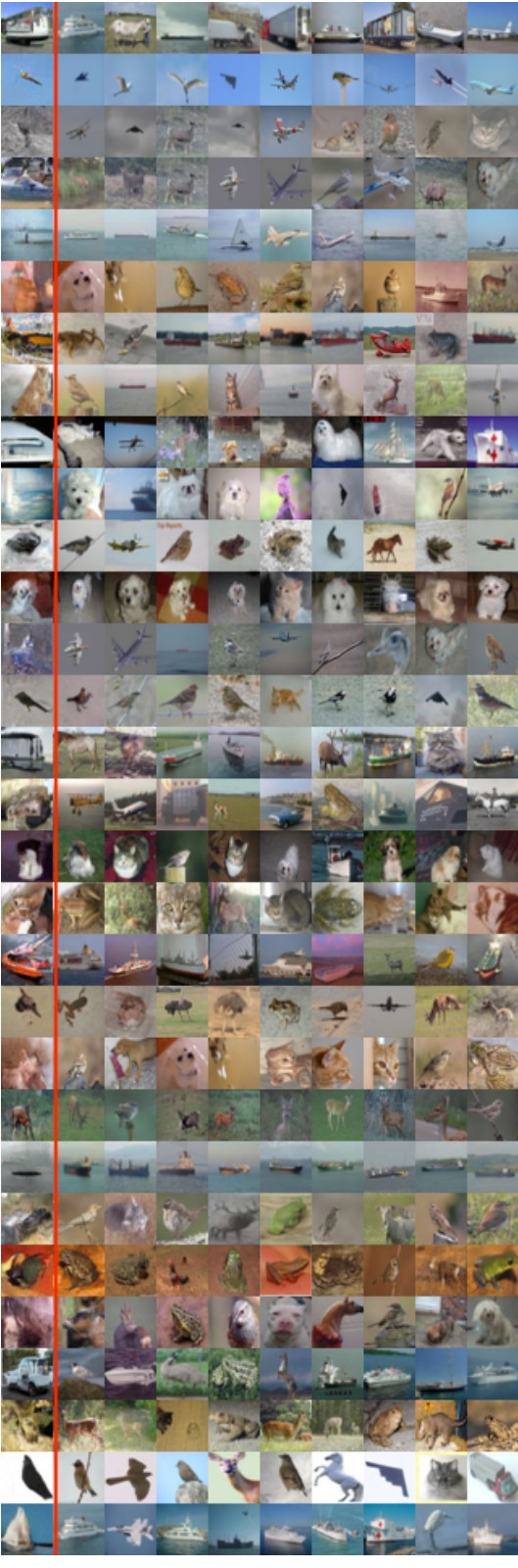

Figure 22: Nearest neighbors measured by the $\ell_2$ distance between images. Images on the left of the red vertical line are samples from our model. Images on the right are nearest neighbors in the training dataset.

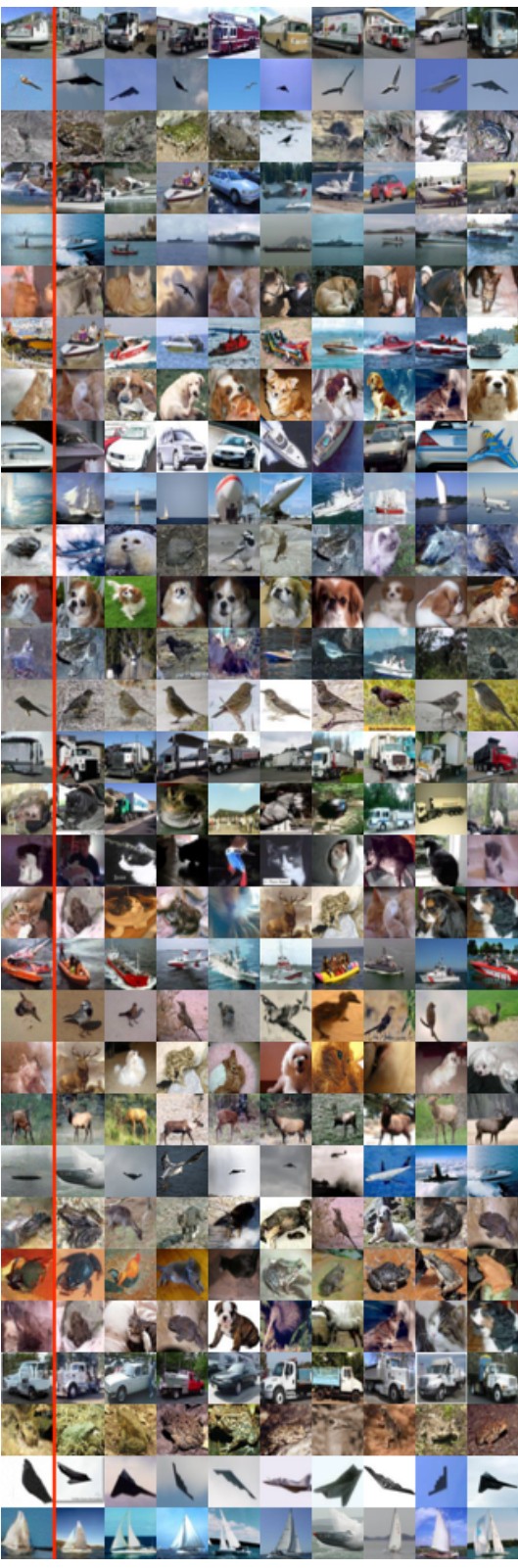

Figure 23: Nearest neighbors measured by the $\ell_2$ distance in the feature space of an Inception V3 network pretrained on ImageNet. Images on the left of the red vertical line are samples from our model. Images on the right are nearest neighbors in the training dataset.

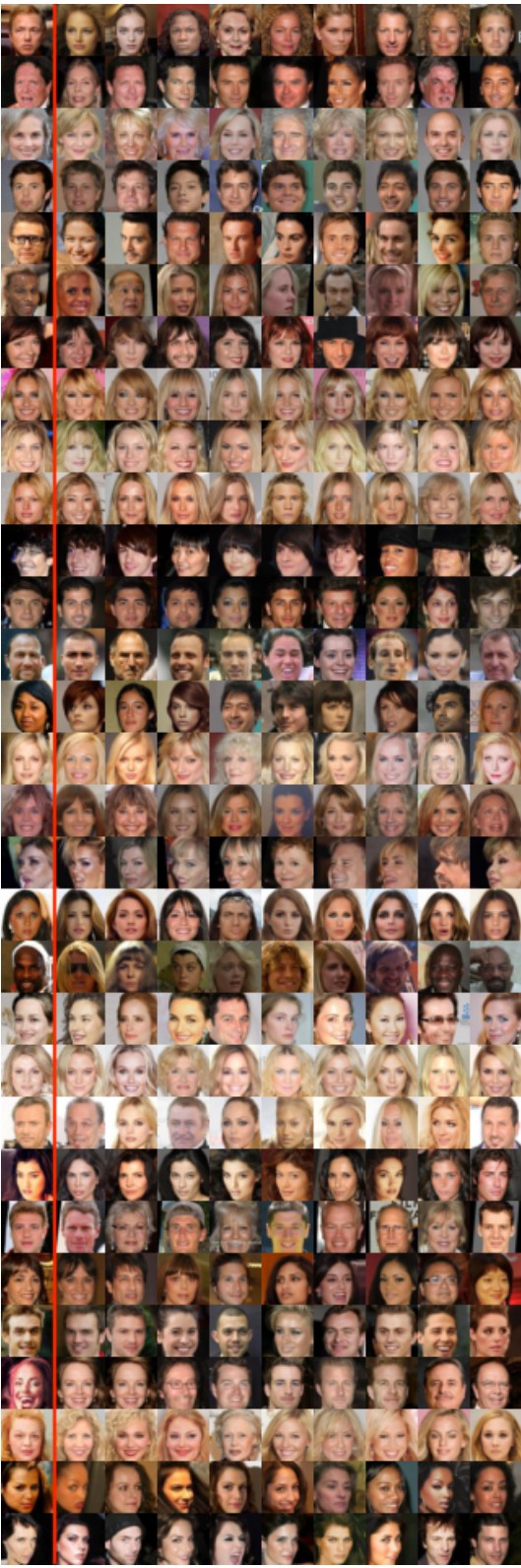

Figure 24: Nearest neighbors measured by the $\ell_2$ distance between images. Images on the left of the red vertical line are samples from our model. Images on the right are nearest neighbors in the training dataset.

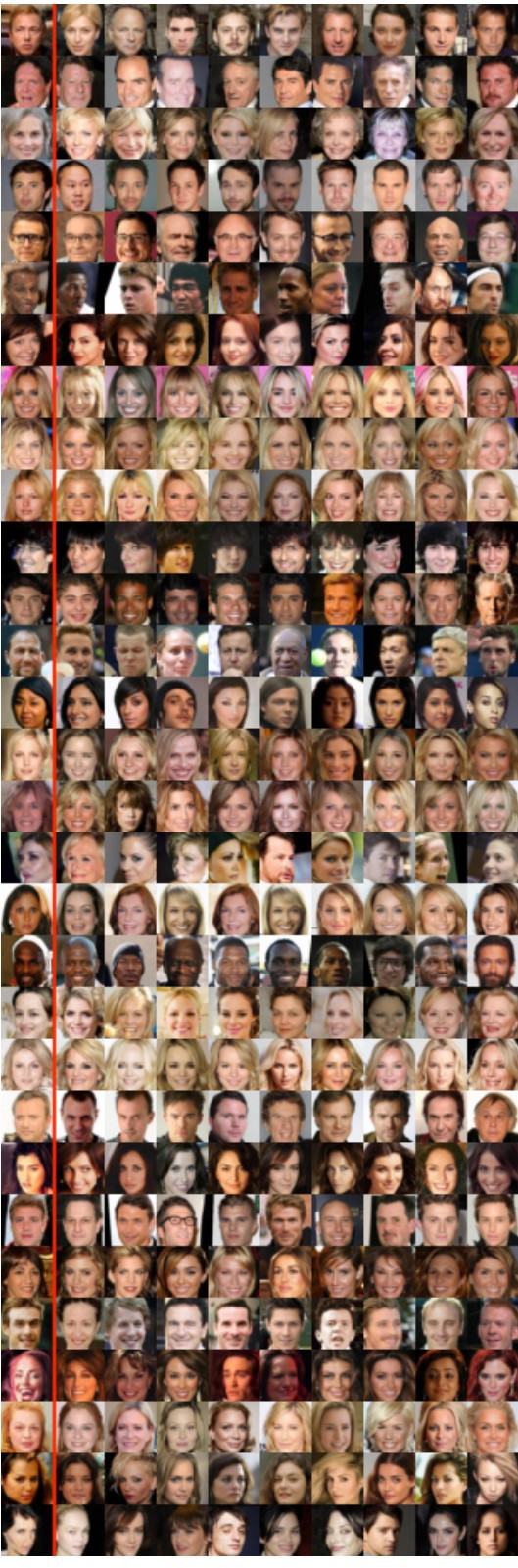

Figure 25: Nearest neighbors measured by the $\ell_2$ distance in the feature space of an Inception V3 network pretrained on ImageNet. Images on the left of the red vertical line are samples from our model. Images on the right are nearest neighbors in the training dataset.

## C.6 ABLATION STUDIES

In this section, we show that gradient-based "single-step denoising" will not improve sample qualities without performing randomized smoothing. To see this, we draw samples from a PixelCNN++ $p_\theta(\mathbf{x})$ trained directly on $p_{\text{data}}(\mathbf{x})$ (*i.e.* without smoothing). We perform "single-step denoising" update defined as

$$\mathbf{x} = \mathbf{x} + \sigma^2 \nabla_{\mathbf{x}} \log p_\theta(\mathbf{x}). \tag{14}$$

We explore various values for $\sigma$, and report the results in figure 26. This shows that "single-step denoising" alone (without randomized smoothing) will not improve sample quality of PixelCNN++.

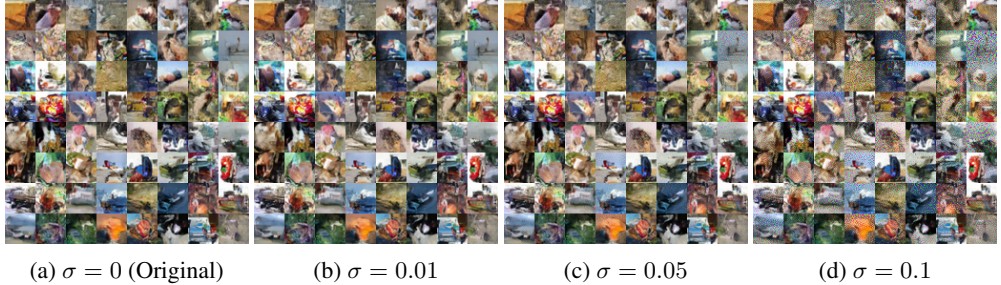

(a) $\sigma = 0$ (Original)    (b) $\sigma = 0.01$    (c) $\sigma = 0.05$    (d) $\sigma = 0.1$

Figure 26: "Single-step denoising" on PixelCNN++ trained on un-smoothed data. $\sigma = 0$ corresponds to the original samples.

