# OpenReview forum: "Improved Autoregressive Modeling with Distribution Smoothing"
_ICLR.cc/2021/Conference — ICLR 2021 Oral_

### Official Review · AnonReviewer4 · 2020-10-24
**Good paper, well motivated and strongly grounded in theory**

**Rating:** 8
**Confidence:** 3

**Review:**

#### Summary of the paper :
The authors propose to improve the sample quality of autoregressive models. The authors propose to (1) - smooth the input data distribution leveraging methods that have shown success in adversarial defense, (2) recover input distribution by learning to reverse the smoothing process. The authors first demonstrate the efficiency of their method on 1d toy-problem, and extend the demonstration to more complex datasets such as MNIST, CIFAR-10 and CelebA with application such as image generation, inpaintting and denoising.

#### Pros :
* The idea to leverage a method previously used for adversarial defense to density estimation is interesting and novel.
* The paper is well motivated through the manifold hypothesis approximation (which results in densities with high Lipschitz constants) and compounding errors.
* The theory is strong

#### Cons :
* The experiments on denoising and inpainting are only qualitative and suffer from a lack of quantitative evaluation.

#### Recommandation :
The article is clear, well motivated, and have a strong theoretical grounding. Therefore I would tend to accept the article.

#### Detailed comments :

* The experiment on 2d synthetic datasets (especially the olympic dataset) should be discussed more thoroughly. First, it is not clear that the proposed model is generating better sample than the MADE baseline on this specific dataset. Second, the intersection between rings, in the olympic dataset, seems to be much poorly modeled with the proposed approach compared to the MADE baseline. What is the reason ?

* In the section 3.2 the authors are introducing 2 different debiasing methods (either a denoising step or another autoregressive model). In the rest of the article it is not clear which of the two methods the authors are using. In addition, in the 2d toy-problem (i.e. ring and olympic) as the authors are choosing a gaussian smoothing both debiasing methods are usable. Therefore it would be interesting to show both methods and to describe thoroughly the differences (in addition, it might provide an answer to my previous point).

* The authors should not mention denoising and inpainting applications if there is no quantitative assessment (at least in appendix)… For the inpainting part, the corrupted input are not even shown (which part of the image has been predicted). The denoising and inpainting experiments sounds like it’s been rushed…

#### Typos and suggestions to improve the paper :
* Minor : Both theorems are provided with nice demonstrations, then the authors should refer to the demonstration in the core text of the article (e.g. see Appendix A).
* Minor : Add small arrows in Table 2 to indicate that Inception score is better when lower, and opposite for FID
* Typo : page 5, section 3.3, paragraph 2 : relative —> relatively
* Figure3 : Right panel : What are the 3 shaded curves ? This should be shown in the legend or at least in the caption
* Figure3 : Right panel: In the x-axis it should be specified ‘Variance of q(x^{\tilde} \mid x)
* Page 7 : paragraph 1 : 'Thus, it is hard to conclusively determine what is the best way of choosing q(x ̃|x).’ —> I  think the authors actually give the key to properly choose the noise level (i.e. variance). It seems to depend on the task : if one wants to generate  good samples, then the variance has to be set by heuristic. If one needs a good likelihood (e.g. for subsequent downstream tasks) then the variance could be optimized.
* Figure 6 :  On my understanding, the part ‘denoising’ is redundant with the section image generation. It is interesting to mention the denoising application, but I am not convinced of the utility of the figure 6.
* Figure 7 : What is the corrupted input ? Which part of the input has been masked ??

---

> ### Author Response · Authors · 2020-11-24
> **Detailed analysis on 2-d synthetic datasets is included. Improved figures in the revision.**
>
> We thank you for the constructive feedback and writing suggestions! We have uploaded the revision to address the feedback.
>
> **Q: Intersection of rings on Olympics datasets and single-step (gradient-based) denoising on 2-d datasets.**
>
> **A:** We believe the reason why the intersections of the rings are not modeled well is because for a smoothed data sample $\tilde x$ near the intersections of the ring, the ground truth $p(x|\tilde x)$ is a complicated distribution that can be hard to be captured by the simple "denoising" model we used, which only has two mixture of logistics at each dimension. If we increase the flexibility of the denoising model, the intersection can be modeled in a better way with our method. We have updated the results in section 4.2 and showed an example with increased mixtures of logistics components for both the baseline (now 6 mixture of logistics, used to be 5) and our methods (now 3 mixture of logistics, used to be 2).
>
> In Appendix B.2, we have added experiments where we compared the performance of our method when using 2, 3 and 4 mixtures of logistics for the MADE model with the baseline MADE model using 5, 6 and 7 mixtures of logistics. We show that by only increasing the mixture components from **2 to 3**, our method can capture the intersections of the Olympics distribution in an improved way. **The baseline MADE model, on the other hand, still has trouble modeling the two sides of the rings when using 7 mixtures of logistics** on the Olympics dataset. This shows the effectiveness of our methods. We have included more analysis in Appendix B.
>
> We also provide single-step denoising results on 2-d datasets in Appendix B.2. However, the denoising results are not very good when the smoothing distribution has a relatively large variance since single-step denoising use $\mathbb {E}[x|\tilde x]$ as a substitute for the denoised $\tilde x$, which might not be ideal in practice.
>
> **Q: Not clear what debiasing methods are used.**
>
> **A:** We have clarified the text and provide “single-step denoising” (gradient based denoising) results in Appendix B.
>
> **Q: The authors should not mention denoising and inpainting applications if there is no quantitative assessment. What is the corrupted input in Figure 7?**
>
> **A:** We have moved the denoising and inpainting experiments to Appendix C.2. In Figure 7 (now Figure 14 in the revision), the bottom half of the image is being inpainted. We have clarified the text and included an image which shows the corrupted input (see Appendix C.2).
>
> **Q: Both theorems are provided with nice demonstrations, the authors should refer to the demonstration in the core text of the article.**
>
> **A:** We have added references to the proofs in the main article.
>
> **Q: Add small arrows in Table 2 and typos in the paper.**
>
> **A:** We have added small arrows to Table 2. We have also fixed the typo in the revision.
>
> **Q: Figure 3 : Right panel : What are the 3 shaded curves? The x-axis should be specified ‘Variance of $q(\tilde x|x)$’.**
>
> **A:** The shaded curves were the loss curves without smoothing the loss values. We have updated the plots (Figure 4 in the revision) by only showing the loss curves after smoothing (the loss values). We have also updated the x-axis according to the feedback.
>
> **Q: It does not seem hard to conclusively determine the best way of choosing $q(\tilde{x}|x)$; if you want good samples, then it has to be set from heuristic; if you want good likelihood, then it can be directly optimized**
>
> **A:** We agree.  By saying  “it is hard to conclusively determine”, we mean it can be hard to design the heuristic that works best in practice.

---

### Official Review · AnonReviewer3 · 2020-10-25
**Advances on autoregressive models. Well-written, simple idea with good results.**

**Rating:** 7
**Confidence:** 3

**Review:**

**Summary.** Autoregressive models have demonstrate their potential utility for modeling images and other types of complex data with high flexibility (particularly in density estimation). However, its sampling ability is not that good as explained in the paper. Authors show that one of the main weaknesses of autoregressive models comes from the propagation of mistakes due to the mismatch of conditionals. Inspired in the promising results of randomized smoothing in adversarial models (Cohen et al. 2019), authors propose a similar strategy. The addition of Gaussian noise and posterior modeling of the smoother data makes easier to the autoregressive density to capture the true data distribution. The benefits of this strategy are empirically proved and shown in the experiments.

**Strengths.** The quality of writing is high and the presentation of the paper facilitates the process of reading. I have to say that I enjoyed while reviewing it. The analysis and description of problems for sampling from autoregressive models is completely understandable to me and I agree with the manifold hypothesis held.

Results with the “sharp” multimodal data looks reliable to me and I believe that the smoother process can also reduce the lipschitz constant as stated in Theorem 1. Until pp. 5, nothing is said about the data denoising process, so one could initially think that there is no way to recover the target density without noise, but authors also did an effort on this. Good point. It is important to remark that the randomized smoothing process can be reverted once learning finishes.

Additionally, I particularly like how authors first present the idea on 1-d examples, later in the experiments, the method is validated with 2-d rings and finally, as stated in the introduction, with different image datasets.

Finally, I did not find any similar work that mixes the idea of smoothing for improving autoregressive modelling.

**Weaknesses, Questions & Recommendations.**
To me, there are 3 main points of weakness:
[W1]. A lack of analysis about the optimal noise for randomized smoothing.
[W2]. Why just Gaussian noise, what if data is discrete, could we do this with another type of noise?
[W3]. Comments about denoising are included a bit late in the manuscript. I think that authors should remark that this is a reversible process.

My main questions are:
[Q1]. In section 2.2, I do not see why data closer to the manifold should have larger first order derivatives or even infinity. Is this a bit counter intuitive, or not? Like, better positioned, worse gradient values?
[Q2]. Is the 1/N term in the global likelihood expression of 1st paragraph of section 2 correct?
[Q3]. If I do not appropriately choose the \sigma parameter for smoothing, do I have the risk of not capturing some modes of the original data? I have the opinion that adding too much or too less noise to data could “mask” modes and something could be lost. Am I correct? Did authors empirically analyzed this in the experiments?
[Q4]. How could we assess that conditionals are now better fitted than before?

A few recommendations for improvement:
[Rec1]. I would explain a bit more the manifold hypothesis of section 2.2, maybe a diagram or figure would help for quicker comprehension of the problem.
[Rec2]. Some acronym for “randomized smoothing” would help in the 1st paragraph of section 3.1. To avoid repetitive expressions.

**Reasons for score.** I liked the idea, think that the paper is well written and I trust the results presented by the authors. Despite the randomized smoothing strategy is rather simple, it seems to work particularly well. For this reason I tend to vote for accept. If I not set a higher score, it is because a bit more of analysis on the optimal sigma, distribution for smoothing and lipschitz constant could have been included.

**Post-rebuttal update.** Thanks to the authors for their response to all my questions and comments. I also read the updated version of the manuscript, which is clearly improved and the rest of reviews and comments by the AC. Looking to that, I agree with the rest of reviewers about the quality of the paper, so I raised my score and I recommend to accept it.

---

> ### Author Response · Authors · 2020-11-24
> **We use the median value of the Euclidean distance between two data points in a dataset for choosing the smoothing distribution on images (Saremi & Hyvarinen, 2019; Garreau et al., 2017).**
>
> We thank Reviewer 3 for the constructive feedback and writing suggestions! We have uploaded the revision to address the feedback.
>
> **Q: A lack of analysis about the optimal noise for randomized smoothing.**
>
> **A:** We discussed the selection of noise in section 3.3. In general, we believe the optimal noise depends on the task. For instance, if our goal is to obtain better likelihoods, optimizing the noise distribution using ELBO could be a way of finding the optimal noise for randomized smoothing. However, if our goal is to generate high quality image samples, training the noise distribution via ELBO might not be the best option since better likelihoods do not necessarily imply better sample qualities (A note on the evaluation of generative models, Theis et al. (2015)). In practice, we also find that optimizing the noise using ELBO is not able to provide high quality image samples.
>
> As there is not an efficient objective function which we can directly optimize on for improved sample quality, we use heuristics from (Saremi & Hyvarinen, 2019; Garreau et al., 2017) and select the noise distribution according to the median value of the Euclidean distance between two data points in a dataset. We find selecting the smoothing distribution with heuristics is able to drastically improve the image sample quality. We believe if there exists efficient objective functions for sample quality just as ELBO for likelihoods, we can directly optimize the objective function to obtain the optimal noise distribution.
>
> **Q: Why just Gaussian noise, what if data is discrete, could we do this with another type of noise?**
>
> **A:** In section 4.1, we study the performance of different types of noise including uniform, Laplace and Gaussian noise on a 1-d toy dataset. We find that in the 1-d example, all the three types of noise are able to work reasonably well, with Gaussian noise slightly outperforming the other two noise distributions. Since state-of-the-art models such as NCSN (Song & Ermon (2019; 2020)) and denoising diffusion probabilistic model (Ho et al., 2020) also use Gaussian noise, this motivates us to use Gaussian noise for the later experiments. We believe the optimal type of noise for other discrete data could be different and domain knowledge may be required to design the optimal noise distribution.
>
> **Q: Comments about denoising are included a bit late in the manuscript.**
>
> **A:** We have included comments about denoising in the introduction.
>
> **Q: In section 2.2, I do not see why data closer to the manifold should have larger first order derivatives or even infinity.**
>
> **A:** We have included an example on the manifold hypothesis in section 2.2. We also add a figure to provide intuition.
>
> **Q: Is the 1/N term in the global likelihood expression of 1st paragraph of section 2 correct?**
>
> **A:** Yes, the 1/N term is correct. It is an unbiased Monte Carlo estimation of the expectation which is equivalent to sampling mean.
>
> **Q: If I do not appropriately choose the sigma parameter for smoothing, do I have the risk of not capturing the original data well?**
>
> **A:** Yes, if the sigma parameter for smoothing is not chosen appropriately, the performance of our method can be affected negatively. The rightmost panel in Figure 3 (Figure 4 in the revision) has demonstrated this point where we compute ELBO for various sigmas on three types of smoothing distributions and we observe a reverse U-shape correlation between the variance of the smoothing distribution and ELBO (see section 4.1 for more details). We also have a related discussion in section 3.3. More specifically, when the smoothing distribution has a zero variance, no noise is added to the data distribution. In this case, modeling the smoothed distribution would be equivalent to modeling $p_{data}(\textbf x)$, which is the same as the original method. When the smoothing distribution has an infinite variance, modeling the denoising model is equivalent to directly modeling $p_{data}(\textbf x)$, which is also the same as the original modeling method.
>
> **Q: How could we assess that conditionals are now better fitted than before?**
>
> **A:** We believe that the visualization and loss curves in the 1-d example in Figure 4 (used to be Figure 3), 2-d example in Figure 5 (used to be Figure 4), and the improved image sample quality in image experiments can provide insights that the model can capture the conditionals reasonably well.
>
> **Q: Some acronyms for “randomized smoothing” in section 3.1**
>
> **A:** We have used “smoothing” as the acronym for “randomized smoothing” in section 3.1 in the revision.

---

### Official Review · AnonReviewer1 · 2020-10-27
**Interesting empirical results but might benefit from more careful discussion.**

**Rating:** 7
**Confidence:** 4

**Review:**

#### SHORT DESCRIPTION
This paper proposes a two-stage generative modeling approach, first learning a distribution over noised data, then learning the original data distribution conditioned on this noised data. The paper demonstrates that this leads to improved sample quality compared to fitting the data distribution directly.


#### DISCUSSION
Overall, I like this paper: it's a straightforward idea, decently motivated and fairly well described, and has good supporting empirical evaluation. I didn't expect the sampling performance to improve substantially by adding just a single denoising step, and I think demonstrating this is a good contribution. However, I think the paper could be improved by some more careful discussion, and a better placement in the literature.

"Theorem 2 shows that our smoothing process provides a regularization effect on the original objective... This regularization effect can, intuitively, increase the generalization capability of the model." How does the extra term in the theorem lead to a regularization effect? Why does this 'intuitively' increase the generalization capability of the model? Unless I'm mistaken, the added term is (up to a constant) the Laplacian of the log-likelihood w.r.t. data. The objective maximizes this on average across observed data, which intuitively minimizes the 'curvature' or 'steepness' of the log-likelihood at observed data, thus presumably smoothing the maximum likelihood solution. This Laplacian term also appears in the score matching objective presented in Theorem 1 of 'Estimation of Non-Normalized Statistical Models by Score Matching, Hyvarinen 2005', where it is minimized instead of maximized. There are also known connections between score matching and denoising methods e.g. 'Optimal Approximation of Signal Priors, Hyvarinen 2006', and 'A Connection Between Score Matching and Denoising Autoencoders, Vincent 2011', which you've cited in passing later. Much of this material and how it relates to the objective in Theorem 2 might be discussed in more depth rather than passing over it as simply a 'regularization term'.

"Our approach is related to two-stage VAE (Dai & Wipf, 2019) which introduces a second VAE to correct the errors made by the first VAE." I'm not sure I agree with this. The idea of the two-stage VAE in that paper is to clean up mismatch between the aggregate posterior q(z) and the prior p(z). On the other hand, your variational model is identical to the canonical VAE setup: x is data, z is noised data, the 'posterior' q(z | x) is fixed and adds Gaussian noise, the 'prior' p(z) is a powerful autoregressive model, and the observation likelihood p(x | z) is another powerful autoregressive model (the canonical VAE would have learned q(z | x), fixed simple p(z), and learned but simple p(x | z)). This is one of the reasons 'VAE' can a confusing term when used to describe latent variable generative models in general: assuming something should be 'encoded' and 'decoded' can sometimes obfuscate the actual probabilistic model. What you propose in this paper might generically be called a 'denoising VAE', but again that's maybe not the most accurate description. I think the most closely related work is probably the denoising diffusion and denoising score-matching approaches which have received attention recently and which you've mentioned, but you could also think of it as turning a denoising autoencoder into a generative model. In any case, I think a more careful discussion of these points would be beneficial for the paper.

Finally, the approach isn't really tied to autoregressive models, apart from the motivation given in terms of smoothing 1D distributions. It's fairly likely that the same idea could readily be applied to e.g. normalizing flows and that it would work well there also, so it would have been nice to see experiments featuring flows included here. This would be especially useful since your best-performing two-stage method takes autoregressive models, which are already slow samplers, and effectively doubles their sampling time.


#### EXTRA NOTES
Maybe be careful with the word 'spurious' in the intro - I know what you mean, but samples from a model are by definition typical samples from that model, and there's nothing spurious about them. They're only questionable when compared to data. Similarly: "The “erroneous” sample xˆ, in some sense, resembles an adversarial example, i.e., an input that causes the model to make mistakes." This seems to be implying that samples generated by a model are somehow pathological. By virtue of the fact that they are generated by the model, they are *by definition* typical samples from the model. There is nothing pathological whatsoever about them. Why the model specification and fitting procedure have resulted in such samples, and whether the samples resemble training data or not, is another issue entirely.

'However, this approach bounds the capacity of the model by limiting the number of modes for each conditional.' All models have limited capacity -- what is particularly bad about the capacity of an autoregressive model being limited in this way? Do we have reason to believe inability to cover multiple nodes is a common bottleneck?

Figure 2 & Figure 3: Axis ticks are too small (and there probably too many), whole figure could be made bigger (this would also help with legends cutting off a lot of the plots).

'Proofs' for theorems 1 and 2 should be referred to in the main text. Theorem 2 should also have log p(x) on the LHS?

Figure 5 caption: "All the samples are not conditioned on class labels." -> None of the samples are conditioned on class labels.

What exactly is being inpainted in Figures 7 (a) and (b)?

Since a central claim of the paper is that the method results in improved sample quality, it might be good to add the Kernel Inception Distance ("Demystifying MMD GANs" Binkowski et al 2018) which has many favourable properties over FID, and is really no more difficult to compute.

#### CONCLUSION
Overall, I think this paper is a nice submission, and would like to see it accepted given a few tweaks.

UPDATE: I've upped my score to a 7, and would like to see the paper accepted.

---

> ### Author Response · Authors · 2020-11-24
> **Our method can also be applied to other models like normalizing flow models.**
>
> We thank Reviewer 1 for the constructive feedback and writing suggestions! We have uploaded the revision to address the feedback.
>
> **Q: How does the extra term in Theorem 2 lead to a regularization effect? Why does this 'intuitively' increase the generalization capability of the model?**
>
> **A:** We have incorporated your feedback into the discussion on Theorem 2 (Proposition 1 in the revision). We have also added its connection to score matching and other denoising methods in the discussion.
>
> **Q: The idea of Two-stage VAE (Dai & Wipf, 2019) might not be directly related to our methods.**
>
> **A:** We agree that the idea and motivation of two-stage VAE are different from ours. We have removed the discussion on two-stage VAE in the revision.
>
> **Q: Applying the same randomized smoothing method to other models like normalizing flow models.**
>
> **A:** Yes, this method is not limited to autoregressive models. It can also be applied to other models like normalizing flow models. In the revision, we introduce section 5, a new section which shows on 2D synthetic datasets that for RealNVP (Dinh et al, (2016)), the “randomized smoothing” approach is also able to generate better samples (according to human observers) than the original method and obtain competitive likelihoods while using comparable number of parameters.
>
> **Q: 'However, this approach bounds the capacity of the model by limiting the number of modes for each conditional.' All models have limited capacity -- what is particularly bad about the capacity of an autoregressive model being limited in this way? Do we have reason to believe inability to cover multiple nodes is a common bottleneck?**
>
> **A:** In Figure 4 (used to be Figure 3), we show on a 1-d example that when the capacity of the model is limited in this way, the model will not perform well if there are more modes in the distribution than that of the model. In this case, the model will assign much higher density to the low density region due to the mode covering property of MLE, which could cause unrealistic samples compared to the data distribution. Even when the model is able to generate enough modes to model the target distribution (see Baseline(8) in Figure 4), the model can still fail to capture the data distribution if the data density has sharp transitions (e.g. a high Lipschitz constant). Since an autoregressive model decomposes a joint distribution into 1-d conditional distributions, the same analysis on this 1-d example could also be applicable to the modeling of each univariate conditional $p(x_i|\textbf x_{<i})$ in autoregressive modeling.
>
> Moreover, autoregressive modeling also suffers from compounding error issues during sampling time, meaning that a failure in capturing $p(x_i|\textbf x_{<i})$ could cause $x_i$ to be sampled from a low density region from $p(x_i|\textbf x_{<i})$, introducing difficulty to sample the later pixels correctly, eventually producing a sample that is unlikely under the data distribution. Empirically, our methods have shown to improve the sample quality of PixelCNN++ by introducing more modes to the univariate conditionals in a flexible way so that the low density regions can also be captured in an improved way, which also aligns with our claims.
>
> **Q:  “Spurious” and “erroneous” are not the right words of describing samples generated by models.**
>
> **A:** We have changed “spurious” and “erroneous”  to “questionable” and “unrealistic” respectively.
>
> **Q: Figure 2 and Figure 3 axis ticks are too small and the whole figure could be made bigger. Figure 5 caption can be improved.**
>
> **A:** We have modified the figures as well as their captions in the revision.
>
> **Q: Proofs for theorems should be referred to in the main text. Theorem 2 should also have log p(x) on the LHS.**
>
> **A:** We have added reference to the proofs of theorems in the main text. Yes, it should be “log p(x)” instead of “p(x)” in Theorem 2, we have fixed the typo in the revision and we thank the reviewer for pointing that out!
>
> **Q: KID scores on image samples.**
>
> **A:**  We evaluated the KID score of our method and compared it to the original PixelCNN++. The mean (standard deviation) KID scores are as follows:
>
> Our method: 0.022 (0.001), the original: 0.046 (0.002).
> The KID score of our model is much better than the original PixelCNN++ baseline.
> As a comparison, according to “Demystifying MMD GANs” (Binkowski et al 2018), WGAN-GP has KID: 0.026 (0.001)
> Cramér GAN has KID: 0.028 (0.001).
> We believe this result has also demonstrated the effectiveness of our method in terms of improving image sample quality.

---

### Official Review · AnonReviewer2 · 2020-10-28
**Good approach. Nice proof-of-concept. Straightforward paper.**

**Rating:** 7
**Confidence:** 3

**Review:**

Summary : This paper proposes an approach to modeling distributions based on a two-step process which involves sampling a noisy x first and then applying a denoising function. The theory is grounded in other works in the literature that use the connection between denoising and the gradient of log p(x) with respect to x.

I enjoyed reading the paper and I think that the authors are definitely working in an exciting area. The authors do a good job in section 3.3 "Tradeoff in modeling" to explain clearly that everything about this "two step" method relies on striking the right balance between the tasks of modeling p(noisy x) and p(x | noisy x). When one is trivial, the other is very hard.

Everything about this paper hinges on the fact that the authors are learning p(noisy x). If they didn't, this whole paper would be trivial and a rather useless exercise. It took me a few passes to realize that they were indeed learning p(noisy x). This is such an important thing, and I think it might be worth insisting a bit more on this. Otherwise it's easy to look at the pictures and to conclude that they are taking x from the training distribution, adding a small amount of noise, and then showing that they can remove the noise. The authors are doing more than that, and this is why I like this paper.


Some more specific comments about the text.

Section 3.3 has "q(noisy x| x) is simply zero" when it's in fact a distribution with all its mass around x; not around zero. We understand what the authors meant by the context, but I recommend rephrasing this.

I like figure 1-2 for how they illustrate the concepts well.

I like that the authors took the time to write Theorem 1 instead of hand waving to appeal to how reasonable result feels.

Theorem 2 contains a typo in its statement. The left side of the equation should be "log p" instead of "p".

I could argue that "Theorem 2" might be more appropriately called simply a "Proposition" instead of a "Theorem", but I will leave it up to the authors to decide that. However, using the \approx notation without clarifying its precise meaning is unbecoming for a theorem. Here the authors intend to communicate something that there is an equality, but with an extra term o(epsilon^2) on the right-hand side, as epsilon->0, with that epsilon having some role in the definition of q(noisy x | x). I think this deserves to be part of the statement of this theorem. Plenty of things are "almost equal" to something else, so the precise meaning matters when stating a theorem.

Figure 5 refers to columns in a way that doesn't feel natural to me. To my eye, column 2 looks bad, column 3 is the best, and column 4 looks the worse. I suspect that the description of the figure either treats the first column as "Column 0", or it starts counting "column 1" after skipping the leftmost column, or we just don't have the same visual appreciation of thumbnails (especially when they contain wild pixels like column 2 does).

Inpainting at the bottom of page 7 doesn't really say what regions of the image are cut out to be inpainted.

Figure 9 says "unconditioned", but it's about p(x | noisy x). Wouldn't that be the opposite of "unconditioned"?

---

> ### Author Response · Authors · 2020-11-24
> **Our smoothing model is unconditioned on class labels during training.**
>
> We would like to first thank Reviewer 2 for the constructive feedback and writing suggestions! We have uploaded the revision to address the feedback.
>
> **Q: "q(noisy x| x) is simply zero" not being precise.**
>
> **A:** We have rephrased "q(noisy x| x) is simply zero" to “a distribution with all its mass at x” in the revision.
>
> **Q: Theorem 2 contains a typo: the left side of the equation should be "log p" instead of "p".**
>
> **A:** Yes, it should be “log p” instead of “p”. We have fixed this typo in Theorem 2 (Proposition 1 in the revision).
>
> **Q: "Theorem 2" might be more appropriately called simply a "Proposition" instead of a "Theorem".**
>
> **A:** We have changed Theorem 2 to Proposition 1, since the core idea is related to the one in (Bishop, 1995).
>
> **Q: Clarification of Figure 5 columns.**
>
> **A:** We have clarified the column number by adding the column number to Figure 5 (Figure 6 in the revision) and updating the caption.
>
> **Q: The regions of images being inpainted.**
>
> **A:** The bottom half of the image is being impainted. We have included text and figures to show the inpainted region. In the revision, we moved the “image inpainting” experiments to Appendix C.2.
>
> **Q: Figure 9 says "unconditioned", but it's about p(x | noisy x). Wouldn't that be the opposite of "unconditioned"?**
>
> **A:** “Unconditioned samples” in Figure 9 (now Figure 17) means the model is not conditioned on class labels. The opposite of "unconditioned" are models conditioned on class labels during training, which would have a positive effect on sample quality. We apologize for the confusion and we have clarified the image captions in the revision.

---

### Comment · ~Qinsheng_Zhang1 · 2021-01-14
**NLL calculation in table 1 and section 5**

Thanks for your insightful paper.

However, I have a small question about facts presented in Section 4.2 2-D synthetic datasets. The paper estimates the density through modeling p(noise x) p(x | noise x), then how can we calculate the exact negative log-likelihoods shown in Table 1?  To calculate p(x), we need to marginalize noise x,  which is intractable by integrating p(noise x, x).


Update: The p(x) is an approximation. There is  q(noise |x) which could be a good proposal distribution for important sampling and also can be approximated by ELBO.

---

> ### Comment · ~Zhisheng_Xiao1 · 2021-01-18
> **I guess it is possible to marginalize in 2-d toy data**
>
> and in high-dim they report ELBO.

---

### Decision · Program_Chairs · 2021-01-07
**Final Decision**

**Decision:**

Accept (Oral)

**Comment:**

All reviewers recommend acceptance. Some concerns were raised about the precision of theorem 2 (now renamed to proposition 1), as well as the analysis of hyperparameter choices and quantitative evaluation, which I believe the authors have adequately addressed. Based on a suggestion of reviewer 1, experiments with flow-based models were also added, which demonstrates that the method is not strictly tied to autoregressive models. Personally, I was also curious about the connection between noise injection and quantisation, which the authors responded to by adding a paragraph discussing this connection in the manuscript.

I would recommend that the authors also add the kernel inception distance (KID) results reported in the comments to the manuscript.

This work stands out to me in that it combines a relatively simple, easy to understand idea with nice results, which is a trait of many impactful papers. I will therefore join the reviewers in recommending acceptance.